# Efficient estimation for large-scale linkage disequilibrium patterns of the human genome

Xin Huang[1,2,3†], Tian-Neng Zhu[1†], Ying-Chao Liu[1], Guo-An Qi[1,4], Jian-Nan Zhang[5], Guo-Bo Chen[2,3,6]*

[1]Institute of Bioinformatics, Zhejiang University, Hangzhou, China; [2]Center for General Practice Medicine, Department of General Practice Medicine, Zhejiang Provincial People's Hospital, People's Hospital of Hangzhou Medical College, Hangzhou, China; [3]Center for Reproductive Medicine, Department of Genetic and Genomic Medicine, and Clinical Research Institute, Zhejiang Provincial People's Hospital, People's Hospital of Hangzhou Medical College, Zhejiang, China; [4]Hainan Institute of Zhejiang University, Hainan, China; [5]Alibaba Group, Hangzhou, China; [6]Key Laboratory of Endocrine Gland Diseases of Zhejiang Province, Hangzhou, China

**\*For correspondence:**
chenguobo@gmail.com

†These authors contributed equally to this work

**Abstract** In this study, we proposed an efficient algorithm (X-LD) for estimating linkage disequilibrium (LD) patterns for a genomic grid, which can be of inter-chromosomal scale or of small segments. Compared with conventional methods, the proposed method was significantly faster, dropped from $O(nm^2)$ to $O(n^2m)$—$n$ the sample size and $m$ the number of SNPs, and consequently we were permitted to explore in depth unknown or reveal long-anticipated LD features of the human genome. Having applied the algorithm for 1000 Genome Project (1KG), we found (1) the extended LD, driven by population structure, universally existed, and the strength of inter-chromosomal LD was about 10% of their respective intra-chromosomal LD in relatively homogeneous cohorts, such as FIN, and to nearly 56% in admixed cohort, such as ASW. (2) After splitting each chromosome into upmost of more than a half million grids, we elucidated the LD of the HLA region was nearly 42 folders higher than chromosome 6 in CEU and 11.58 in ASW; on chromosome 11, we observed that the LD of its centromere was nearly 94.05 folders higher than chromosome 11 in YRI and 42.73 in ASW. (3) We uncovered the long-anticipated inversely proportional linear relationship between the length of a chromosome and the strength of chromosomal LD, and their Pearson's correlation was on average over 0.80 for 26 1KG cohorts. However, this linear norm was so far perturbed by chromosome 11 given its more completely sequenced centromere region. Uniquely chromosome 8 of ASW was found most deviated from the linear norm than any other autosomes. The proposed algorithm has been realized in C++ (called X-LD) and is available at https://github.com/gc5k/gear2, and can be applied to explore LD features in any sequenced populations.

## eLife assessment

This study presents a **useful** new approach for efficient computation of statistics on correlations between genetic variants (linkage disequilibrium, or LD), which the authors apply to quantify the extent of LD across chromosomes. The method and its derivation are **solid**. The authors document that cross-chromosome LD can be substantial, which has implications for geneticists who are interested in population structure and its impact on genetic association studies.

## Introduction

Linkage disequilibrium (LD) is the association for a pair of loci and the metric of LD serves as the basis for developing genetic applications in agriculture, evolutionary biology, and biomedical research (*Weir, 2008*; *Hill and Robertson, 1966*). The structure of LD of the human genome is shaped by many factors, mutation, recombination, population demography, epistatic fitness, and completeness of genomic data itself (*Myers et al., 2005*; *Nei and Li, 1973*; *Ardlie et al., 2002*). Due to its overwhelming cost, LD structure investigation is often compromised to a small genomic region (*Chang et al., 2015*; *Theodoris et al., 2021*), and their typical LD structure is as illustrated for a small segment (*Barrett et al., 2005*). Now, given the availability of large-scale genomic data, such as millions of single-nucleotide polymorphisms (SNPs), the large-scale LD patterns of the human genome play crucial roles in determining genomics studies, and many theories and useful algorithms upon large-scale LD structure, from genome-wide association studies, polygenic risk prediction for complex diseases, and choice for reference panels for genotype imputation (*Vilhjálmsson et al., 2015*; *Yang and Zhou, 2020*; *Bulik-Sullivan et al., 2015*; *Yang et al., 2011*; *Das et al., 2016*).

However, there are impediments, largely due to intensified computational cost, in both investigating large-scale LD and providing high-resolution illustrations for their details. If we consider a genomic grid that consists of $m^2$ SNP pairs, given a sample of $n$ individuals and $m$ SNPs ($n \ll m$)—typically as observed in 1000 Genomes Project (1KG) (*Lowy et al., 2019*), its benchmark computational time cost for calculating all pairwise LD is $\mathcal{O}\left(nm^2\right)$, a burden that quickly drains computational resources given the volume of the genomic data. In practice, it is of interest to know the mean LD of the $m_i^2$ SNP pairs for a genomic grid, which covers $m_i \times m_j$ SNP pairs. Upon how a genomic grid is defined, a genomic grid consequently can consist of (1) the whole genome-wide $m^2$ SNP pairs, and we denote their mean LD as $\ell_g$ ; (2) the intra-chromosomal mean LD for the $i$th chromosome of $m_i^2$ SNP pairs, and denote as $\ell_i$ ; and (3) the inter-chromosomal mean LD $i$th and $j$th chromosomal $m_i m_j$ SNP pairs, and denoted as $\ell_{i\cdot j}$ .

In this study, we propose an efficient algorithm that can estimate $\ell_g$ , $\ell_i$ , and $\ell_{i\cdot j}$ , the computational time of which can be reduced from $\mathcal{O}\left(nm_i^2\right)$ to $\mathcal{O}\left(n^2 m_i\right)$ for $\ell_i$ and $\mathcal{O}\left(nm_i m_j\right)$ to $\mathcal{O}\left(n^2 m_i + n^2 m_j\right)$ for $\ell_{i\cdot j}$ . The rationale of the proposed method relies on the connection between the genetic relationship matrix (GRM) and LD (*Chen, 2014*; *Goddard, 2009*), and in this study a more general transformation

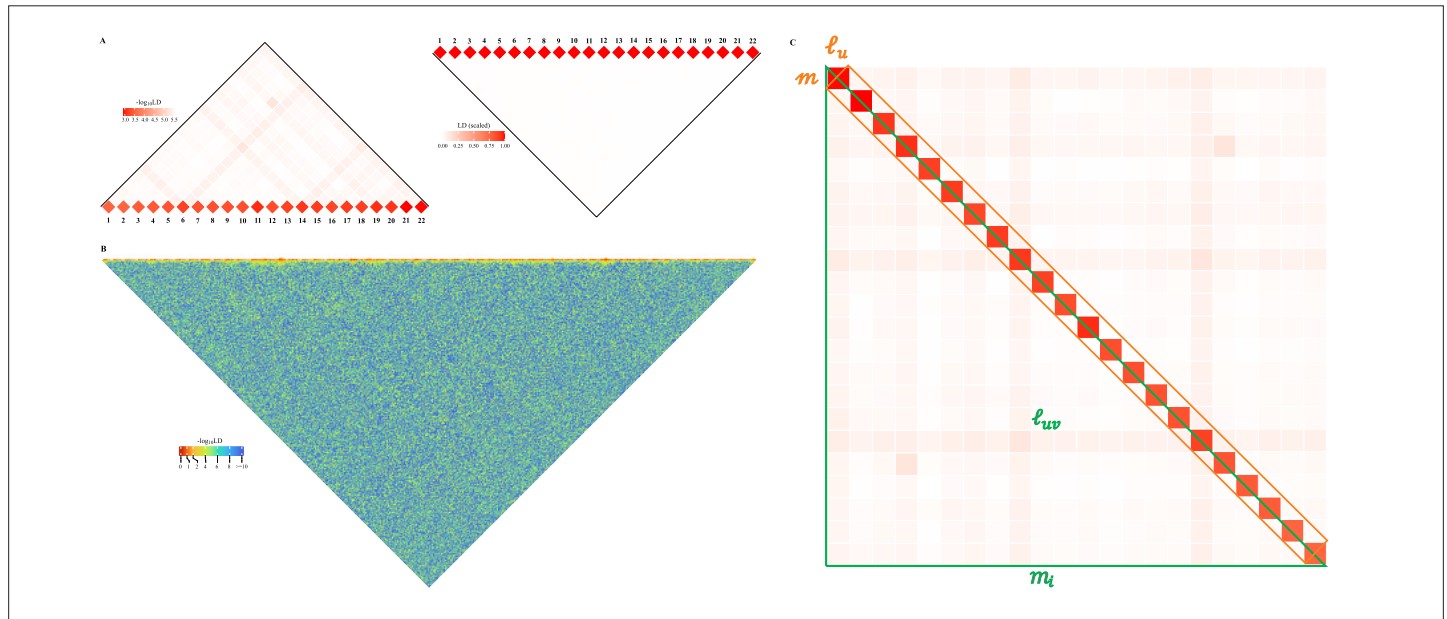

**Figure 1.** Schematic illustration for large-scale linkage disequilibrium (LD) analysis as exampled for CONVERGE cohort. (**A**) The 22 human autosomes have consequently 22 $\hat{\ell}_i$ and 231 $\hat{\ell}_{i\cdot j}$ , without (left) and with (right) scaling transformation; Scaling transformation is given in *Equation 8*. (**B**) If zoom into chromosome 2 of 420,946 single-nucleotide polymorphisms (SNPs), a chromosome of relative neutrality is expected to have self-similarity structure that harbors many approximately strong $\hat{\ell}_u$ along the diagonal, and relatively weak $\hat{\ell}_{uv}$ off-diagonally. Here chromosome 2 of CONVERGE has been split into 1000 blocks and yielded 1000 $\hat{\ell}_u$ LD grids, and 499,500 $\hat{\ell}_{uv}$ LD grids. (**C**) An illustration of the construction process for the LD-decay regression model.

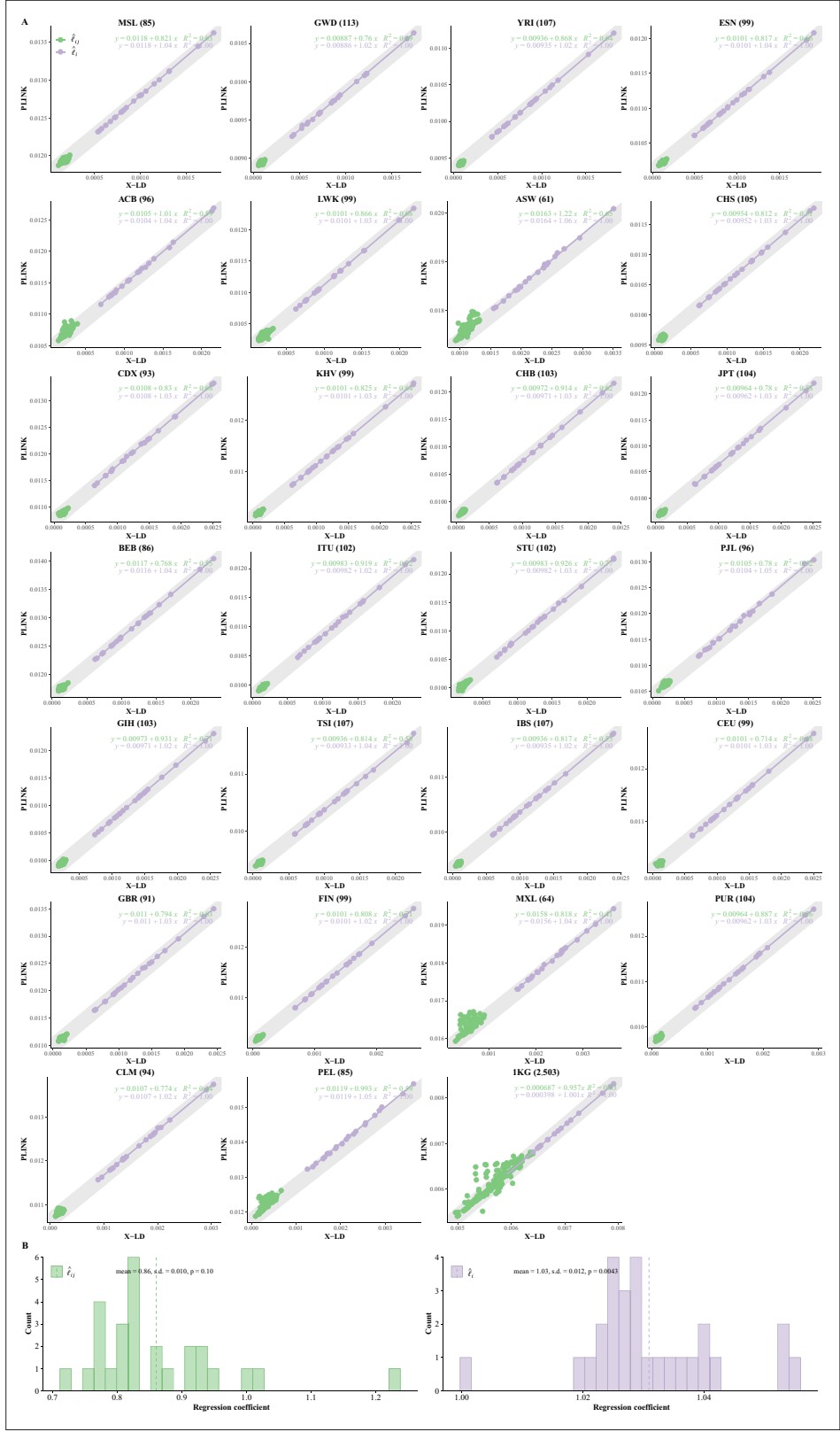

**Figure 2.** Reconciliation for linkage disequilibrium (LD) estimators in the 26 cohorts of 1KG. (**A**) Consistency examination for the 26 1KG cohorts for their $\hat{\ell}_i$ and $\hat{\ell}_{i\cdot j}$ estimated by X-LD and PLINK (--r2). In each figure, the 22 $\hat{\ell}_i$ fitting line is in purple, whereas the 231 $\hat{\ell}_{i\cdot j}$ fitting line is in green. The gray solid line, $y = \frac{1}{n} + x$, in which $n$ the sample size of each cohort, represents the expected fit between PLINK and X-LD estimates, and the two estimated

*Figure 2 continued on next page*

*Figure 2 continued*

regression models at the top-right corner of each plot show this consistency. The sample size of each cohort is in parentheses. (**B**) Distribution of $R^2$ of $\hat{\ell}_i$ and $\hat{\ell}_{i \cdot j}$ fitting lines is based on X-LD and PLINK algorithms in the 26 cohorts; $R^2$ represents variation explained by the fitted model. 26 1KG cohorts: MSL (Mende in Sierra Leone), GWD (Gambian in Western Division, The Gambia), YRI (Yoruba in Ibadan, Nigeria), ESN (Esan in Nigeria), ACB (African Caribbean in Barbados), LWK (Luhya in Webuye, Kenya), ASW (African Ancestry in Southwest US), CHS (Han Chinese South), CDX (Chinese Dai in Xishuangbanna, China), KHV (Kinh in Ho Chi Minh City, Vietnam), CHB (Han Chinese in Beijing, China), JPT (Japanese in Tokyo, Japan), BEB (Bengali in Bangladesh), ITU (Indian Telugu in the UK), STU (Sri Lankan Tamil in the UK), PJL (Punjabi in Lahore, Pakistan), GIH (Gujarati Indian in Houston, TX), TSI (Toscani in Italia), IBS (Iberian populations in Spain), CEU (Utah residents [CEPH] with Northern and Western European ancestry), GBR (British in England and Scotland), FIN (Finnish in Finland); MXL (Mexican Ancestry in Los Angeles, CA), PUR (Puerto Rican in Puerto Rico), CLM (Colombian in Medellin, Colombia), and PEL (Peruvian in Lima, Peru).

The online version of this article includes the following figure supplement(s) for figure 2:

**Figure supplement 1.** Reconciliation for linkage disequilibrium (LD) estimators in AFR, EAS, and EUR.

**Figure supplement 2.** The computational efficiency of X-LD algorithm.

from GRM to LD can be established via Isserlis's theorem (*Isserlis, 1918*; *Zhou, 2017*). The statistical properties, such as sampling variance, of the estimated LD have been derived too.

The proposed method can be analogously considered a more powerful realization for Haploview (*Barrett et al., 2005*), but additional utility can be derived to bring out an unprecedented survey of LD patterns of the human genome. As demonstrated in 1KG, we consequently investigate how biological factors such as population structure, admixture, or variable local recombination rates can shape large-scale LD patterns of the human genomes.

1. The proposed method provides statistically unbiased estimates for large-scale LD patterns and shows computational merits compared with the conventional methods (Figure 2).
2. We estimated $\ell_g$ and 22 autosomal $\ell_i$ and 231 inter-autosomal $\ell_{i \cdot j}$ for the 1KG cohorts. There was a ubiquitous existence of extended LD, which was associated with population structure or admixture (Figure 3).
3. We provided high-resolution illustration that decomposed a chromosome into upmost nearly a million grids, each of which was consisted of 250 × 250 SNP pairs, the highest resolution that has been realized so far at autosomal level (Figure 4); tremendous variable recombination rates led to regional strong LD as highlighted for the HLA region of chromosomes 6 and the centromere region of chromosome 11.
4. Furthermore, a consequently linear regression constructed could quantify LD decay score genome-widely, and in contrast LD decay was previously surrogated in a computationally expensive method. There was a strong ethnicity effect that was associated with extended LD (Figure 5).
5. We demonstrate that the strength of autosomal $\ell_i$ was inversely proportional to the SNP number, an anticipated relationship that is consistent with genome-wide spread of recombination hotspots. However, chromosome 8 of ASW showed substantial deviation from the fitted linear relationship (Figure 6).

The proposed algorithm has been realized in C++ and is available at https://github.com/gc5k/gear2, (copy archived at *Chen, 2023*). As tested, the software could handle sample sizes as large as 10,000 individuals.

## Methods
### The overall rationale for large-scale LD analysis

We assume LD for a pair of biallelic loci is measured by squared Pearson's correlation, $\rho^2_{l_1 l_2} = \frac{D^2_{l_1 l_2}}{p_{l_1} q_{l_1} p_{l_2} q_{l_2}}$, in which $D_{l_1 l_2}$ the LD of loci $l_1$ and $l_2$, $p.$ and $q.$ the reference and the alternative allele frequencies. If we consider the averaged LD for a genomic grid over $m_i^2$ SNP pairs, the conventional estimator is $\hat{\ell}_i = \frac{1}{m_i^2} \sum_{l_1, l_2}^{m_i} \rho^2_{l_1 l_2}$, and, if we consider the averaged LD for $m_i$ and $m_j$ SNP pairs between two genomic segments, then $\hat{\ell}_{i \cdot j} = \frac{1}{m_i m_j} \sum_{l_1, l_2}^{m_i, m_j} \rho^2_{l_1 l_2}$. Now let us consider the 22 human autosomes (*Figure 1A*).

We naturally partition the genome into $\mathcal{C} = 22$ blocks, and its genomic LD, denoted as $\ell_g$, can be expressed as

$$\ell_g = \frac{1}{m^2}\sum_{l_1,l_2}^{m}\rho_{l_1l_2}^2 = \frac{1}{m^2}\left(\sum_{i}^{e}\left(\sum_{l_1,l_2}^{m_i}\rho_{l_1l_2}^2\right) + \sum_{i\neq j}^{e}\left(\sum_{l_1}^{m_i}\sum_{l_2}^{m_j}\rho_{l_1l_2}^2\right)\right) = \sum_{i}^{e}\frac{m_i^2}{m^2}\ell_i + \sum_{i\neq j}^{e}\frac{m_im_j}{m^2}\ell_{i\cdot j} \qquad (1)$$

So we can decompose $\ell_g$ into $\mathcal{C}\ell_i$ and $\frac{\mathcal{C}(\mathcal{C}-1)}{2}$ unique $\ell_{i\cdot j}$. Obviously, **Equation 1** can be also expressed in the context of a single chromosome $\ell_i = \frac{1}{\beta_i^2}\left(\sum_{u}^{\beta_i}\ell_u + \sum_{u\neq v}^{\beta_i}\ell_{uv}\right)$, in which $\beta_i = \frac{m_i}{m}$ the number of SNP segments, each of which has $m$ SNPs. Geometrically it leads to $\beta_i$ diagonal grids and $\frac{\beta_i(\beta_i-1)}{2}$ unique off-diagonal grids (**Figure 1B**).

**Table 1.** Notation definitions.

| Notation | Definition |
|---|---|
| $\mathcal{C}$ | The number of chromosomes. |
| $i$ and $j$ | Subscripts index chromosome $i$ and $j$. |
| $\beta_i$ | The number of SNP segments of chromosome $i$, each of which has $\mathfrak{m}$ SNPs. |
| $D_{l_1l_2}$ | The difference between the observed and expected haplotype frequencies, with $D_{l_1l_2} = p_{l_1l_2} - p_{l_1}p_{l_2}$. |
| $F$ | The inbreeding coefficient. |
| $\mathbf{K}_i$ | Genetic relatedness matrix for chromosome $i$, and two vectors, $k_{i_o}$ and $k_{i_d}$, from $\mathbf{K}_i$, where $k_{i_o}$ stacks the off-diagonal elements and $k_{i_d}$ stacks the diagonal elements. |
| $k$ | Subscript indexes individual. |
| $l_1$ and $l_2$ | Subscripts index a pair of SNPs. |
| $m$ | The number of SNPs; $m_i$ the number of SNPs on chromosome $i$. |
| $n$ | The number of samples; $n_i$, the number of samples in subpopulation $i$. |
| $p_l$ and $q_l$ | Frequency of the $l$th reference allele and alternative allele in the population. |
| $\theta_{k_1k_2}$ | The relatedness score between individual $k_1$ and $k_2$. |
| $x_{kl}$ | The genotype for the $k$th individual at the $l$th biallelic locus. |
| $\mathbf{X}_i$ and $\widetilde{\mathbf{X}}_i$ | Genotype and standardized genotype matrixes for chromosome $i$. |
| $\rho_{l_1l_2}^2$ | Squared Pearson's correlation coefficient for any pair of SNPs, including an SNP to itself when $l_1 = l_2$. |
| $r^2$ | Squared Pearson's correlation metric for LD but estimated from PLINK (--r2) or PopLDdecay. |
| $\ell_g$ | The mean LD of the whole genome-wide $m^2$ SNP pairs. |
| $\ell_i$ | The intra-chromosomal mean LD for the $i$th chromosome of $m_i^2$ SNP pairs. |
| $\ell_{i\cdot j}$ | The inter-chromosomal mean LD $i$th and $j$th chromosomal $m_im_j$ SNP pairs, a scaled version is $\widetilde{\ell}_{ij}$. |
| $\ell_u$ | The mean LD for a diagonal grid. |
| $\ell_{uv}$ | The mean LD for off-diagonal grids. |

LD, linkage disequilibrium; SNP, single-nucleotide polymorphism.

**Table 2.** Computational time for the demonstrated estimation tasks.

| Cohort | Task description | Time cost | Computational time complex |
|---|---|---|---|
| CHB ($n = 103, m = 2,997,655$) | Estimation for 22 autosomal $\ell_i$, and 231 inter-chromosomal $\ell_{i \cdot j}$. For results, see **Figure 3** and **Table 3**. | 101,34 s | $\mathcal{O}\left(n^2 m\right)$ |
| 1KG ($n = 2,503, m = 2,997,655$) | Same as above. | 3008.29 s | Same as above |
| CONVERGE ($n = 10,640$, $m = 5,215,820$) | Same as above. For results, see **Figure 1A**. | 77,508.00 s | Same as above |
| | Estimation for high-resolution LD interaction given bin size of 250 SNPs | | |
| CHB ($n = 103, m_2 = 241,241$) | Chromosome 2, estimation for 965 $\ell_i$, and 465,130 $\ell_{i \cdot j}$. For results, see **Figure 4**. | 66.86 s | $\mathcal{O}\left(n^2\left(m_i + \left(\frac{m_i}{250}\right)^2\right)\right)$ |
| CHB ($n = 103, m_{22} = 40,378$) | Chromosome 22, estimation for 162 $\ell_i$, and 13,041 $\ell_{i \cdot j}$. For results, see **Figure 4**. | 3.22 s | Same as above |
| CONVERGE ($n = 10,640$, $m_{22} = 71,407$) | Chromosome 22, estimation for 286 $\ell_i$, and 40,755 $\ell_{i \cdot j}$. | 8,736.29 s | Same as above |
| CONVERGE ($n = 10,640$, $m_2 = 420,949$) | Chromosome 2, estimation for 1000 $\ell_i$, and 499,500 $\ell_{i \cdot j}$. For results, see **Figure 1B**. | 45,125.00 s | Chromosome 2 was split into 1000 blocks, each of which had about 420 SNPs |

For the sake of fair comparison, 10 CPUs were used for multi-thread computing.

LD, linkage disequilibrium; SNP, single-nucleotide polymorphism.

## LD-decay regression

As human genome can be boiled down to small LD blocks by genome-widely spread recombination hotspots (**Hinch et al., 2019**; **Li et al., 2022**), mechanically there is self-similarity for each chromosome that the relatively strong $\ell_i$ for juxtaposed grids along the diagonal but weak $\ell_{i \cdot j}$ for grids slightly off-diagonal. So, for a chromosomal $\ell_i$, we can further express it as

$$\ell_i = \frac{1}{\beta_i^2}\left(\sum_u^{\beta_i} \ell_u + \sum_{u \neq v}^{\beta_i} \ell_{uv}\right) = E\left(\ell_u\right)\frac{1}{\beta_i} + E\left(\ell_{uv}\right)\left(1 - \frac{1}{\beta_i}\right) = \frac{1}{\beta_i}\left[E\left(\ell_u\right) - E\left(\ell_{uv}\right)\right] + E\left(\ell_{uv}\right) \quad (2)$$

in which $\ell_u$ is the mean LD for a diagonal grid, $\ell_{uv}$ the mean LD for off-diagonal grids, and $m_i$ the number of SNPs on the $i$th chromosome. Consider a linear model below (see **Figure 1C** for its illustration),

$$\ell = b_0 + b_1 x + e \quad (3)$$

in which $\ell$ represents a vector composed of $\mathcal{C}$ $\ell_i, x$, $x$ represents a vector composed of $\mathcal{C} x_i$, and $x_i = \frac{1}{m_i}$ the inversion of the SNP number of the $i$th chromosome. The regression coefficient and intercept can be estimated as below:

$$b_1 = \frac{cov\left(x, \ell\right)}{var\left(x\right)} = \frac{E\left(x\ell\right) - E\left(x\right)E\left(\ell\right)}{var\left(x\right)}$$

and

$$b_0 = E\left(\ell\right) - b_1 E\left(x\right)$$

There are some technical details in order to find the interpretation for $b_0$ and $b_1$. We itemize them briefly. For the mean and variance of $x$:

$$\begin{cases} E\left(x\right) = \frac{1}{\mathcal{C}}\sum_i^{\mathcal{C}} \frac{1}{m_i} \\ var\left(x\right) = \frac{1}{\mathcal{C}}\sum_i^{\mathcal{C}} \frac{1}{m_i^2} - \left(\frac{1}{\mathcal{C}}\sum_i^{\mathcal{C}} \frac{1}{m_i}\right)^2 \end{cases}$$

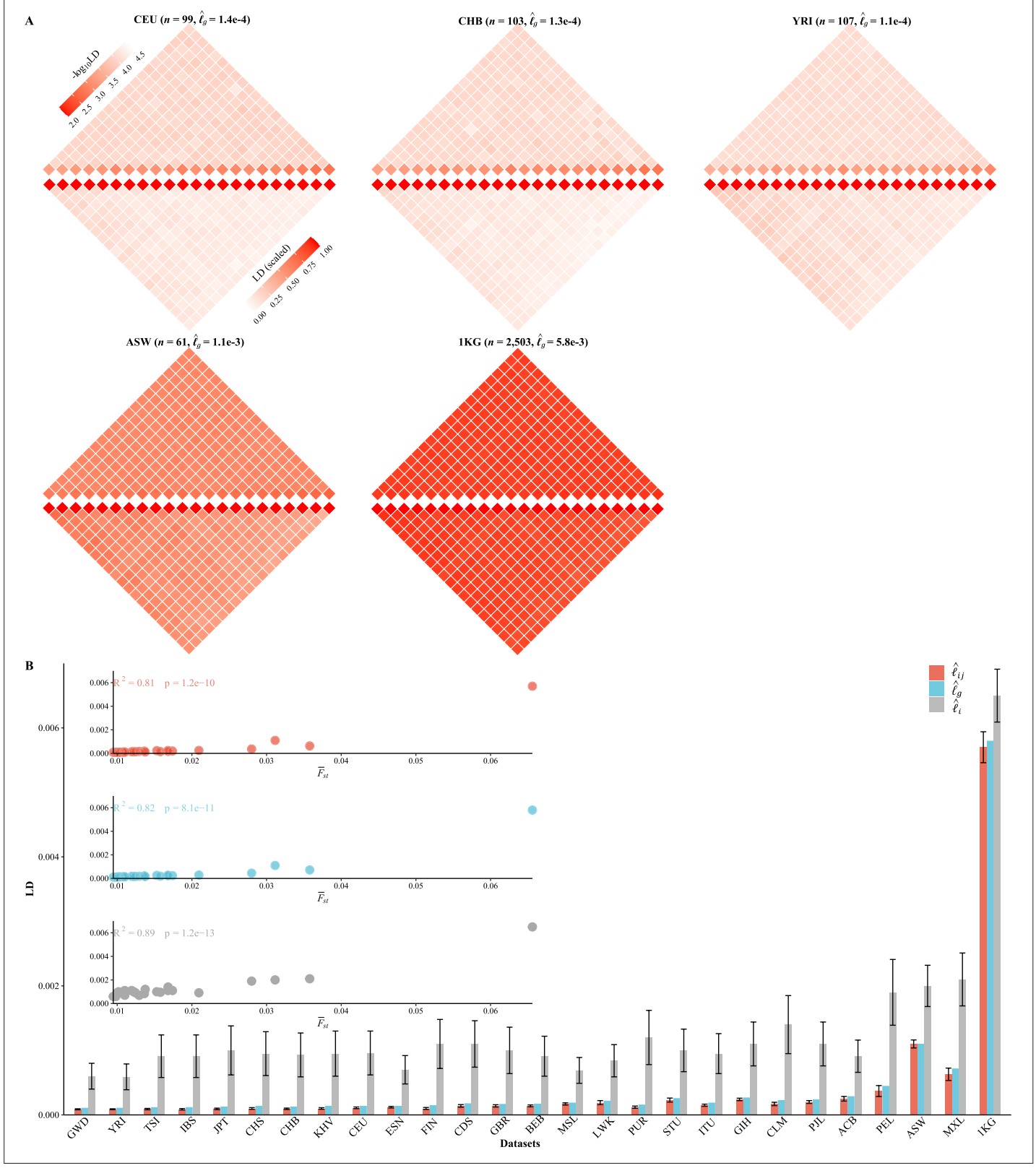

**Figure 3.** Various linkage disequilibrium (LD) components for the 26 1KG cohorts. (**A**) Chromosomal scale LD components for five representative cohorts (CEU, CHB, YRI, ASW, and 1KG). The upper parts of each figure represent $\hat{\ell}_i$ (along the diagonal) and $\hat{\ell}_{i \cdot j}$ (off-diagonal), and the lower part $\tilde{\hat{\ell}}_{i \cdot j}$ as in *Equation 8*. For visualization purposes, the quantity of LD before scaling is transformed to a -log10 scale, with smaller values (red hues) representing

*Figure 3 continued on next page*

*Figure 3 continued*

larger LD, and a value of 0 representing that all single-nucleotide polymorphisms (SNPs) are in LD. (**B**) The relationship between the degree of population structure (approximated by $F_{st}$) and $\hat{\ell}_i$ , $\hat{\ell}_g$ , and $\hat{\ell}_{i \cdot j}$ in the 26 1KG cohorts.

The online version of this article includes the following figure supplement(s) for figure 3:

**Figure supplement 1.** Chromosomal scale linkage disequilibrium (LD) components for 26 cohorts in 1KG.

For $E\left(x\ell\right)$:

$$E(x\ell) = \frac{\sum_i^e \frac{1}{m_i} \left\{ E(\ell_u \frac{m}{m_i}) + E(\ell_{uv}) \left(1 - \frac{m}{m_i}\right) \right\}}{e}$$

$$= \frac{\sum_i^e \left\{ E(\ell_u \frac{m}{m_i^2}) + E(\ell_{uv}) \left(1 - \frac{m}{m_i}\right) \right\}}{e}$$

$$= \left[ (E\left(\ell_u\right) - E\left(\ell_{uv}\right)) m \right] \left( \frac{1}{e} \sum_i^e \frac{1}{m_i^2} \right) + E\left(\ell_{uv}\right) \left( \frac{1}{e} \sum_i^e \frac{1}{m_i} \right)$$

For $E\left(x\right) E\left(\ell\right)$:

$$E\left(x\right) E\left(\ell\right) = \left\{ \frac{1}{e} \sum_i^e \frac{1}{m_i} \right\} \left\{ \left( \frac{1}{e} \sum_i^e \frac{1}{m_i} \right) \left( E\left(\ell_u\right) \cdot m \right) + \left[ 1 - m \left( \frac{1}{e} \sum_i^e \frac{1}{m_i} \right) \right] E\left(\ell_{uv}\right) \right\}$$

$$= \left[ (E\left(\ell_u\right) - E\left(\ell_{uv}\right)) m \right] \left( \frac{1}{e} \sum_i^e \frac{1}{m_i} \right)^2 + E\left(\ell_{uv}\right) \left( \frac{1}{e} \sum_i^e \frac{1}{m_i} \right)$$

Then we integrate these items to have the expectation for $b_1$ :

$$E\left(b_1\right) = \frac{E\left(x\ell\right) - E\left(x\right) E\left(\ell\right)}{var\left(x\right)}$$

$$= \frac{\left\{ \left[ (E\left(\ell_u\right) - E\left(\ell_{uv}\right)) m \right] \left( \frac{1}{e} \sum_i^e \frac{1}{m_i^2} \right) + E\left(\ell_{uv}\right) \left( \frac{1}{e} \sum_i^e \frac{1}{m_i} \right) \right\} - \left\{ \left[ (E\left(\ell_u\right) - E\left(\ell_{uv}\right)) m \right] \left( \frac{1}{e} \sum_i^e \frac{1}{m_i} \right)^2 + E\left(\ell_{uv}\right) \left( \frac{1}{e} \sum_i^e \frac{1}{m_i} \right) \right\}}{\frac{1}{e} \sum_i^e \frac{1}{m_i^2} - \left( \frac{1}{e} \sum_i^e \frac{1}{m_i} \right)^2}$$

$$= \left[ E\left(\ell_u\right) - E\left(\ell_{uv}\right) \right] m$$

Similarly, we plug in $E\left(b_1\right)$ so as to derive $b_0$ :

$$E\left(b_0\right) = E\left(\ell\right) - E\left(b_1\right) E\left(x\right)$$

$$= \left\{ \left( \frac{1}{e} \sum_i^e \frac{1}{m_i} \right) \left( E\left(\ell_u\right) \cdot m \right) + \left[ 1 - m \left( \frac{1}{e} \sum_i^e \frac{1}{m_i} \right) \right] E\left(\ell_{uv}\right) \right\}$$

$$- \left\{ \left( E\left(\ell_u\right) - E\left(\ell_{uv}\right) \right) \cdot m \cdot \left( \frac{1}{e} \sum_i^e \frac{1}{m_i} \right) \right\} = E\left(\ell_{uv}\right)$$

After some algebra, if $E\left(\ell_u\right) \gg E\left(\ell_{uv}\right)$—say if the former is far greater than the latter, the interpretation of $b_1$ and $b_0$ can be

$$\begin{cases} E\left(b_1\right) = E\left(\ell_u - \ell_{uv}\right) m \approx E\left(\ell_u\right) m \\ E\left(b_0\right) = E\left(\ell_{uv}\right) \end{cases} \tag{4}$$

It should be noticed that $E\left(b_1\right) \approx E\left(\ell_u\right) m$ quantifies the averaged LD decay of the genome. Conventional LD decay is analyzed via the well-known LD decay analysis, but *Equation 4* provides a direct estimate of both LD decay and possible existence of extended LD. We will see the application of the model in Figure 5 that the strength of the long-distance LD is associated with population structure. Of note, the underlying assumption of *Equations 3* and *4* is genome-wide spread of recombination hotspots, an established result that has been revealed and confirmed (*Hinch et al., 2019*).

**Table 3.** X-LD estimation for complex LD components (2,997,635 SNPs).

| Cohort ($n$) | Ancestry | $\lambda_1$ ($\hat{F}_{st}$)* | $\hat{\ell}_g$ (SE)† | $\hat{\ell}_i$ (SD)‡ | $\overline{\hat{\ell}_{ij}}$ (SD)‡ | $\tilde{\ell}_{i\cdot j}$ (SD)‡ | Lower bound of LD § |
|---|---|---|---|---|---|---|---|
| MSL (85) | AFR | 1.10 (0.013) | 1.9e-4 (1.21e-6) | 6.9e-4 (2.0e-4) | 1.7e-4 (1.7e-5) | 0.26 (0.053) | 0.16197183 |
| GWD (113) | AFR | 1.07 (0.009) | 1.1e-4 (5.61e-7) | 6.0e-4 (2.0e-4) | 8.7e-5 (8.1e-6) | 0.16 (0.037) | 0.247218789 |
| YRI (107) | AFR | 1.05 (0.010) | 1.1e-4 (4.23e-7) | 5.9e-4 (2.0e-4) | 8.8e-5 (6.9e-6) | 0.16 (0.04) | 0.242001641 |
| ESN (99) | AFR | 1.09 (0.011) | 1.4e-4 (7.67e-7) | 7.0e-4 (2.2e-4) | 1.2e-4 (1.2e-5) | 0.19 (0.043) | 0.217391304 |
| ACB (96) | AFR | 2.01 (0.021) | 2.9e-4 (3.78e-6) | 9.1e-4 (2.5e-4) | 2.5e-4 (3.6e-5) | 0.29 (0.070) | 0.147727273 |
| LWK (99) | AFR | 1.35 (0.014) | 2.2e-4 (2.38e-6) | 8.4e-4 (2.5e-4) | 1.9e-4 (3.2e-5) | 0.24 (0.052) | 0.173913043 |
| ASW (61) | AFR | 1.90 (0.031) | 1.1e-3 (2.73e-5) | 2.0e-3 (3.2e-4) | 1.1e-3 (6.2e-5) | 0.57 (0.059) | 0.079681275 |
| CHS (105) | EA | 1.08 (0.010) | 1.4e-4 (9.39e-7) | 9.5e-4 (3.4e-4) | 1.0e-4 (1.3e-5) | 0.12 (0.030) | 0.31147541 |
| CDX (93) | EA | 1.11 (0.012) | 1.8e-4 (1.38e-6) | 1.1e-3 (3.6e-4) | 1.4e-4 (2.0e-5) | 0.14 (0.040) | 0.272277228 |
| KHV (99) | EA | 1.07 (0.011) | 1.4e-4 (7.67e-7) | 9.5e-4 (3.5e-4) | 1.0e-4 (1.2e-5) | 0.12 (0.031) | 0.31147541 |
| CHB (103) | EA | 1.07 (0.010) | 1.3e-4 (6.94e-7) | 9.3e-4 (3.4e-4) | 9.5e-5 (1.1e-5) | 0.11 (0.030) | 0.317948718 |
| JPT (104) | EA | 1.06 (0.010) | 1.3e-4 (7.22e-7) | 1.0e-3 (3.8e-4) | 9.3e-5 (1.2e-5) | 0.10 (0.028) | 0.338638673 |
| BEB (86) | SA | 1.07 (0.012) | 1.7e-4 (8.09e-7) | 9.1e-4 (3.1e-4) | 1.4e-4 (1.5e-5) | 0.17 (0.042) | 0.236363636 |
| ITU (102) | SA | 1.61 (0.016) | 1.9e-4 (1.84e-6) | 9.5e-4 (3.1e-4) | 1.5e-4 (1.7e-5) | 0.18 (0.044) | 0.231707317 |
| STU (102) | SA | 1.56 (0.015) | 2.6e-4 (3.21e-6) | 1.0e-3 (3.3e-4) | 2.3e-4 (3.1e-5) | 0.23 (0.047) | 0.171526587 |
| PJL (96) | SA | 1.67 (0.017) | 2.4e-4 (2.74e-6) | 1.1e-3 (3.4e-4) | 2.0e-4 (2.2e-5) | 0.21 (0.048) | 0.20754717 |
| GIH (103) | SA | 1.73 (0.017) | 2.7e-4 (3.41e-6) | 1.1e-3 (3.4e-4) | 2.4e-4 (1.9e-5) | 0.23 (0.049) | 0.179153094 |
| TSI (107) | EUR | 1.07 (0.010) | 1.2e-4 (6.10e-7) | 9.1e-4 (3.3e-4) | 9.0e-5 (1.1e-5) | 0.11 (0.029) | 0.325 |
| IBS (107) | EUR | 1.07 (0.010) | 1.2e-4 (6.10e-7) | 9.1e-4 (3.3e-4) | 8.8e-5 (1.1e-5) | 0.11 (0.028) | 0.329949239 |
| CEU (99) | EUR | 1.07 (0.011) | 1.4e-4 (7.67e-7) | 9.6e-4 (3.4e-4) | 1.1e-4 (1.3e-5) | 0.12 (0.030) | 0.293577982 |
| GBR (91) | EUR | 1.11 (0.012) | 1.7e-4 (1.08e-6) | 1.0e-3 (3.6e-4) | 1.4e-4 (1.8e-5) | 0.15 (0.036) | 0.255807107 |
| FIN (99) | EUR | 1.09 (0.011) | 1.5e-4 (9.69e-7) | 1.1e-3 (3.8e-4) | 1.0e-4 (1.5e-5) | 0.10 (0.027) | 0.34375 |
| MXL (64) | AMR | 2.29 (0.036) | 7.2e-4 (1.49e-5) | 2.1e-3 (4.1e-4) | 6.3e-4 (9.6e-5) | 0.32 (0.072) | 0.136986301 |
| PUR (104) | AMR | 1.43 (0.014) | 1.6e-4 (1.30e-6) | 1.2e-3 (4.2e-4) | 1.2e-4 (1.7e-5) | 0.11 (0.026) | 0.322580645 |
| CLM (94) | AMR | 1.58 (0.017) | 2.3e-4 (2.49e-6) | 1.4e-3 (4.5e-4) | 1.7e-4 (2.6e-5) | 0.13 (0.035) | 0.281690141 |
| PEL (85) | AMR | 2.38 (0.028) | 4.5e-4 (7.33e-6) | 1.9e-3 (5.1e-4) | 3.7e-4 (8.5e-5) | 0.21 (0.062) | 0.196483971 |
| 1KG (2503) | MIX | 164.20 (0.066) | 5.8e-3 (4.63e-6) | 6.5e-3 (4.1e-4) | 5.7e-3 (2.4e-4) | 0.88 (0.028) | 0.051505547 |

LD, linkage disequilibrium; SNPs, single-nucleotide polymorphisms.

*Eigenvalue was estimated. In parentheses is the ratio between the listed largest eigenvalue and the sample size. Since there exists an approximation that $\overline{F}_{st} \approx \frac{\lambda_1}{n}$, the ratio can be taken as an approximation of population structure.

†Standard error was calculated as $\sqrt{\frac{2}{n(n-1)}}\left[\hat{\ell}_g - \frac{1}{(n-1)^2}\right]$, as **Equation 7**.

‡Estimated empirically from $\mathcal{C}$ chromosomal $\hat{\ell}_i$; Estimated empirically from $\frac{\mathcal{C}(\mathcal{C}-1)}{2}$ inter-chromosomal $\hat{\ell}_{ij}$.

§It is estimated by $\frac{22\hat{\ell}_i}{22\hat{\ell}_i + 231\hat{\ell}_{ij}}$, indicating lower bound of true LD.

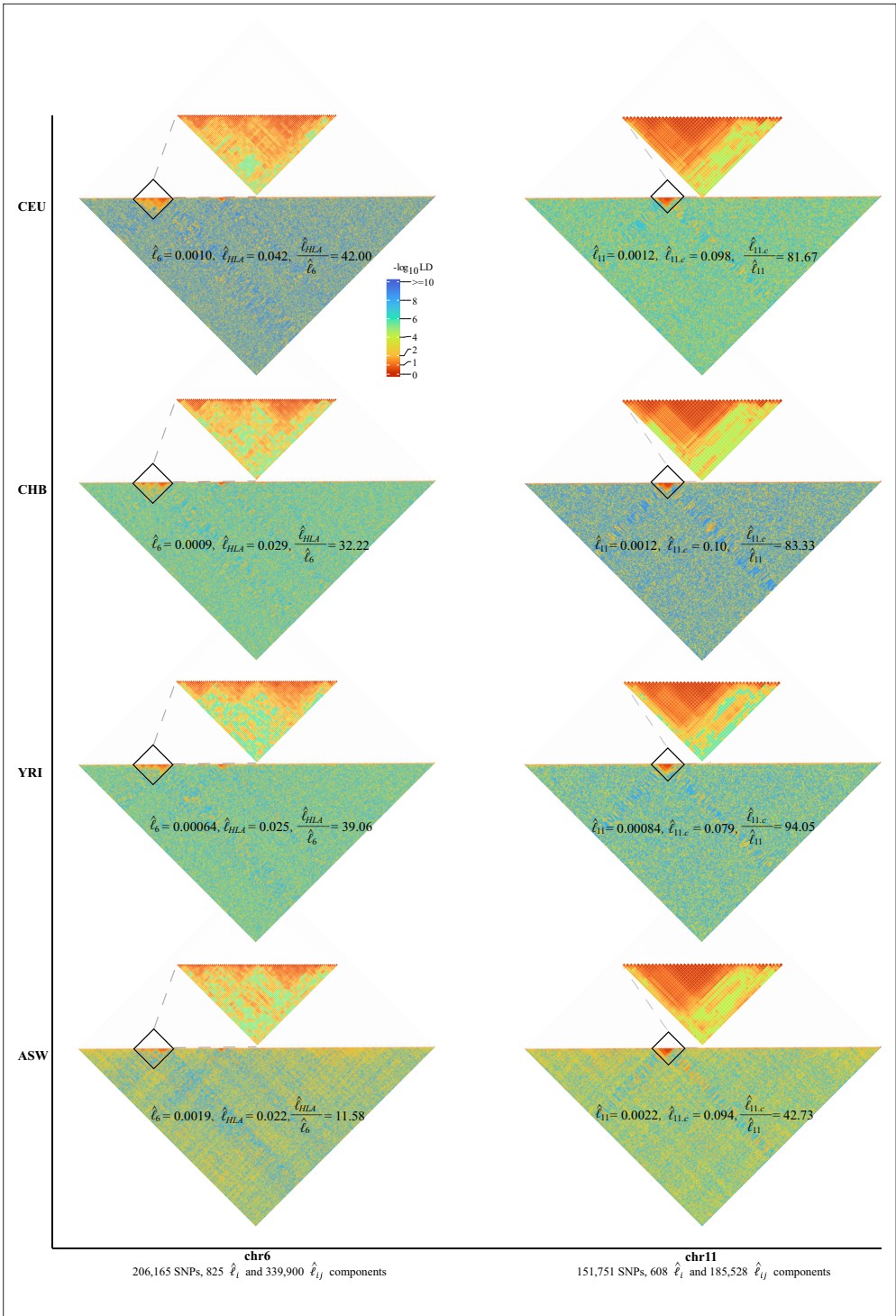

**Figure 4.** High-resolution illustration for linkage disequilibrium (LD) grids for CEU, CHB, YRI, and ASW ($\mathfrak{m} = 250$). For each cohort, we partition chromosomes 6 and 11 into high-resolution LD grids (each LD grid contains 250 ×250 single-nucleotide polymorphism [SNP] pairs). The bottom half of each figure shows the LD grids for the entire chromosome. Further zooming into HLA on chromosome 6 and the centromere region on chromosome 11, and their detailed LD in the relevant regions are also provided in the upper half of each figure. For visualization purposes, LD is transformed to a -log10-scale, with smaller values (red hues) representing larger LD, and a value of 0 representing that all SNPs are in LD.

The online version of this article includes the following figure supplement(s) for figure 4:

*Figure 4 continued on next page*

*Figure 4 continued*

**Figure supplement 1.** High-resolution illustration for linkage disequilibrium (LD) grids for CEU, CHB, YRI, and ASW ($\mathfrak{m} = 500$).

**Figure supplement 2.** Influence of HLA region on chromosome 6 and centromere region on chromosome 11 on chromosomal linkage disequilibrium (LD) in CEU, CHB, YRI, and ASW.

## Efficient estimation for $\ell_g$, $\ell_i$, and $\ell_{i \cdot j}$

For the aforementioned analyses, the bottleneck obviously lies in the computational cost in estimating $\ell_i$ and $\ell_{i \cdot j}$. $\ell_i$ and $\ell_{i \cdot j}$ are used to be estimated via the current benchmark algorithm as implemented in PLINK (*Chang et al., 2015*), and the computational time complex is proportional to $\mathcal{O}\left(nm^2\right)$. We present a novel approach to estimate $\ell_i$ and $\ell_{i \cdot j}$. Given a genotypic matrix $\mathbf{X}$, a $n \times m$ matrix, if we assume that there are $m_i$ and $m_j$ SNPs on chromosomes $i$ and $j$, respectively, we can construct $n \times n$ genetic relatedness matrices as below:

$$
\begin{cases}
\mathbf{K}_i = \frac{1}{m_i} \tilde{\mathbf{X}}_i \tilde{\mathbf{X}}_i^{\mathrm{T}} \\
\mathbf{K}_j = \frac{1}{m_j} \tilde{\mathbf{X}}_j \tilde{\mathbf{X}}_j^{\mathrm{T}}
\end{cases}
\tag{5}
$$

in which $\widetilde{\mathbf{X}}_i$ is the standardized $\mathbf{X}_i$ and $\widetilde{x}_{kl} = \frac{x_{kl} - 2p_l}{\sqrt{2(1+F)p_l q_l}}$, where $x_{kl}$ is the genotype for the $k$th individual at the $l$th biallelic locus, $F$ is the inbreeding coefficient having the value of 0 for random mating population and 1 for an inbred population, and $p_l$ and $q_l$ are the frequencies of the reference and the alternative alleles ($p_l + q_l = 1$), respectively. When GRM is given, we can obtain some statistical characters of $\mathbf{K}_i$. We extract two vectors $\boldsymbol{k}_{i_o}$, which stacks the off-diagonal elements of $\mathbf{K}_i$, and $\boldsymbol{k}_{i_d}$, which takes the diagonal elements of $\mathbf{K}_i$. The mathematical expectation of $\boldsymbol{k}_{i_o}^2$, in which $E(\boldsymbol{k}_{i_o}^2) = \frac{1}{n(n-1)} \sum_{k_1 \neq k_2}^{n} k_{k_1, k_2}^2$, can be established according to Isserlis's theorem in terms of the four-order moment (*Isserlis, 1918*),

$$
E(\boldsymbol{k}_{i_o}^2) = \frac{1}{m_i^2 n(n-1)} \sum_{k_1 \neq k_2}^{n} \sum_{l_1, l_2}^{m_i} \left[ \left(1 + \theta_{k_1 k_2}^2\right) \rho_{l_1 l_2}^2 + \theta_{k_1 k_2}^2 \right]
\tag{6}
$$

in which $E(\theta_{k_1 k_2}) = \left(\frac{1}{2}\right)^r$ is the expected relatedness score and $r$ indicates the $r$th-degree relatives. $r = 0$ for the same individual, and $r = 1$ for the first-degree relatives. Similarly, we can derive for $E\left(\boldsymbol{k}_{i_o} \boldsymbol{k}_{j_o}\right)$. *Equation 6* establishes the connection between GRM and the aggregated LD estimation that $\ell_i = E\left(\boldsymbol{k}_{i_o}^2\right)$. According to Delta method as exampled in Appendix I of *Lynch and Walsh, 1998*, the means and the sampling variances for $\ell_i$ and $\ell_{i \cdot j}$ are

$$
\begin{cases}
E(\boldsymbol{k}_{i_o}^2) = \ell_i = \frac{1}{m_i^2} \sum_{l_1, l_2}^{m_i} \rho_{l_1, l_2}^2 \\
var(\ell_i) = \frac{4[\widehat{var}(\boldsymbol{k}_{i_o})]^2}{n(n-1)} \\
E(\boldsymbol{k}_{i_o} \boldsymbol{k}_{k_o}) = \ell_{i \cdot j} = \frac{1}{m_i m_j} \sum_{l_1, l_2 = 1}^{m_i, m_j} \rho_{l_1, l_2}^2 \\
var\left(\boldsymbol{k}_{i_o}\right) = E\left(\boldsymbol{k}_{i_o}^2\right) - \left[E\left(\boldsymbol{k}_{i_o}\right)\right]^2 = \ell_i - \frac{1}{(n-1)^2}
\end{cases}
\tag{7}
$$

in which $var(\boldsymbol{k}_{i_o}) = E(\boldsymbol{k}_{i_o}^2) - \left[E\left(\boldsymbol{k}_{i_o}\right)\right]^2 = \ell_i - \frac{1}{(n-1)^2}$ and $cov\left(\boldsymbol{k}_{i_o}, \boldsymbol{k}_{j_o}\right) = E\left(\boldsymbol{k}_{i_o} \boldsymbol{k}_{j_o}\right) - E\left(\boldsymbol{k}_{i_o}\right) E\left(\boldsymbol{k}_{j_o}\right) = \ell_{i \cdot j} - \frac{1}{(n-1)^2}$, respectively. Of note, the properties of $\ell_g$ can be derived similarly if we replace $\ell_i$ with $\ell_g$ in *Equation 7*. We can develop $\widetilde{\ell}_{i \cdot j}$, a scaled version of $\ell_{i \cdot j}$, as below:

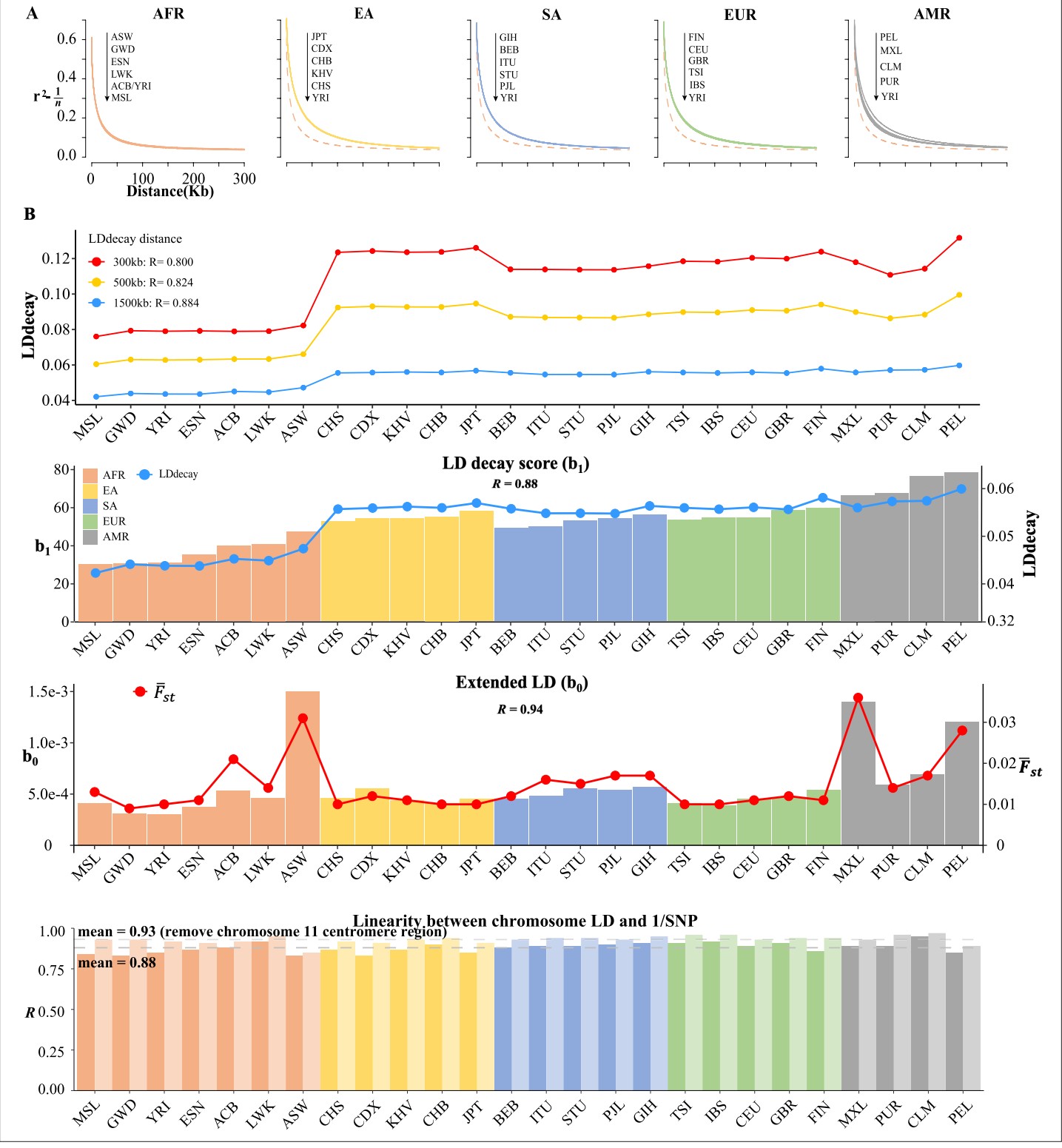

**Figure 5.** Linkage disequilibrium (LD) decay analysis for 26 1KG cohorts. (**A**) Conventional LD decay analysis in PLINK for 26 cohorts. To eliminate the influence of sample size, the inverse of sample size has been subtracted from the original LD values. The YRI cohort, represented by the orange dotted line, is chosen as the reference cohort in each plot. The top-down arrow shows the order of LDdecay values according to Table 5. (**B**) Model-based LD decay analysis for the 26 1KG cohorts. We regressed each autosomal $\hat{\ell}_i$ against its corresponding inversion of the single-nucleotide polymorphism (SNP) number for each cohort. Regression coefficient $b_1$ quantifies the averaged LD decay of the genome and intercept $b_0$ provides a direct estimate of the possible existence of long-distance LD. The $\mathcal{R}$ values in the first three plots indicate the correlation between $b_1$ and LD decay score in three different

*Figure 5 continued on next page*

*Figure 5 continued*

physical distance and the correlation between $b_1$ (left-side vertical axis) and LD decay score (right-side vertical axis) and the correlation between $b_0$ (left-side vertical axis) and $\bar{F}_{st}$ (right-side vertical axis), respectively. The last plot assessed the impact of centromere region of chromosome 11 on the linear relationship between chromosomal LD and the inverse of the SNP number. The dark and light gray dashed lines represent the mean of the $\mathcal{R}$ with and without the presence of centromere region of chromosome 11.

The online version of this article includes the following figure supplement(s) for figure 5:

**Figure supplement 1.** The correlation between the inverse of the single-nucleotide polymorphism (SNP) number and chromosomal linkage disequilibrium (LD) in 26 cohorts of 1KG.

$$\widetilde{\ell}_{i \cdot j} = \frac{\ell_{i \cdot j}}{\sqrt{\widetilde{\ell}_i \widetilde{\ell}_j}} \tag{8}$$

in which $\widetilde{\ell}_i = \frac{m_i \ell_i - 1}{m_i - 1}$ , a modification that removed the LD with itself. According to Delta method, the sampling variance of $\widetilde{\ell}_{i \cdot j}$ is

$$var(\widetilde{\ell}_{i \cdot j}) = \frac{2 \left( \widehat{\ell}_{i \cdot j} \right)^2}{n(n-1)} \left[ \frac{\widehat{var}(\mathbf{k}_{i_o}) \widehat{var}(\mathbf{k}_{j_o})}{\left( \widehat{cov}(\mathbf{k}_{i_o}, \mathbf{k}_{j_o}) \right)^2} + \frac{\left( \widehat{cov}(\mathbf{k}_{i_o}, \mathbf{k}_{j_o}) \right)^2}{\widehat{var}(\mathbf{k}_{i_o}) \widehat{var}(\mathbf{k}_{j_o})} - 2 \right] \tag{9}$$

Of note, when there is no LD between a pair of loci, $\ell$ yields zero and its counterpart PLINK estimate yields $\frac{1}{n}$ , a difference that can be reconciled in practice (see **Figure 2**).

## Raise of LD due to population structure

In this study, the connection between LD and population structure is bridged via two pathways below, in terms of a pair of loci and of the aggregated LD for all pair of loci. For a pair of loci, their LD is often simplified as $\rho^2_{l_1 l_2} = \frac{D^2_{l_1 l_2}}{p_{l_1} q_{l_1} p_{l_2} q_{l_2}}$ , but will be inflated if there are subgroups (**Nei and Li, 1973**). In addition, it is well established the connection between population structure and eigenvalues, and in particular the largest eigenvalue is associated with divergence of subgroups (**Patterson et al., 2006**). In this study, the existence of subgroups of cohort is surrogated by the largest eigenvalue $\lambda_1$ or $\bar{F}_{st} \approx \frac{\lambda_1}{n}$ .

## Data description and quality control

The 1KG (**Auton et al., 2015**), which was launched to produce a deep catalog of human genomic variation by whole-genome sequencing (WGS) or whole-exome sequencing (WES), and 2503 strategically selected individuals of global diversity were included (containing 26 cohorts). We used the following criteria for SNP inclusion for each of the 26 1KG cohorts: (1) autosomal SNPs only; (2) SNPs with missing genotype rates higher than 0.2 were removed, and missing genotypes were imputed; and (3) only SNPs with minor allele frequencies higher than 0.05 were retained. Then 2,997,635 consensus SNPs that were present in each of the 26 cohorts were retained. According to their origins, the 26 cohorts are grouped as African (AFR: MSL, GWD, YRI, ESN, ACB, LWK, and ASW), European (EUR: TSI, IBS, CEU, GBR, and FIN), East Asian (EA: CHS, CDX, KHV, CHB, and JPT), South Asian (SA: BEB, ITU, STU, PJL, and GIH), and American (AMR: MXL, PUR, CLM, and PEL), respectively.

In addition, to test the capacity of the developed software (X-LD), we also included CONVERGE cohort ($n = 10,640$), which was used to investigate major depressive disorder (MDD) in the Han Chinese population (**Cai et al., 2015**). We performed the same criteria for SNP inclusion as that of the 1KG cohorts, and $m = 5,215,820$ SNPs remained for analyses.

## X-LD software implementation

The proposed algorithm has been realized in our X-LD software, which is written in C++ and reads in binary genotype data as often used in PLINK. As multi-thread programming is adopted, the efficiency of X-LD can be improved upon the availability of computational resources. We have tested X-LD in various independent datasets for its reliability and robustness. Certain data management options, such as flexible inclusion or exclusion of chromosomes, have been built into the commands of X-LD. In X-LD, missing genotypes are naively imputed according to Hardy–Weinberg proportions; however,

**Table 4.** Estimates for 22 autosomal $\hat{\ell}_i$ in CEU, CHB, YRI, and ASW, respectively.

| Chromosome | SNP number | $\hat{\ell}_i$ | | | |
| --- | --- | --- | --- | --- | --- |
| | | CEU | CHB | YRI | ASW |
| 1 | 225,967 | 5.0e-4 (8.2e-6) | 0.00049 (7.8e-6) | 0.00032 (4.3e-6) | 0.0015 (4e-05) |
| 2 | 241,241 | 5.0e-4 (8.1e-6) | 5.0e-4 (7.9e-6) | 3.0e-4 (4.1e-6) | 0.0015 (4e-05) |
| 3 | 212,670 | 6.0e-04 (1.0e-5) | 0.00058 (9.5e-6) | 0.00039 (5.7e-6) | 0.0018 (5.1e-5) |
| 4 | 222,241 | 0.00062 (1.0e-5) | 0.00061 (1.0e-5) | 0.00038 (5.4e-6) | 0.0018 (5.0e-5) |
| 5 | 193,632 | 0.00069 (1.2e-5) | 7.0e-04 (1.2e-5) | 0.00043 (6.5e-6) | 0.0018 (4.9e-5) |
| 6 | 206,165 | 0.0010 (1.9e-5) | 9.0e-04 (1.6e-5) | 0.00064 (1.0e-5) | 0.0019 (5.4e-5) |
| 7 | 177,414 | 0.00073 (1.3e-5) | 0.00071 (1.2e-5) | 0.00045 (6.8e-6) | 0.0016 (4.3e-5) |
| 8 | 163,436 | 0.00075 (1.3e-5) | 0.00069 (1.2e-5) | 0.00043 (6.5e-6) | 0.0022 (6.4e-5) |
| 9 | 129,440 | 0.00074 (1.3e-5) | 0.00074 (1.3e-5) | 0.00047 (7.2e-6) | 0.0018 (5.0e-5) |
| 10 | 152,251 | 0.00078 (1.4e-5) | 8.0e-04 (1.4e-5) | 0.00058 (9.3e-6) | 0.0019 (5.6e-5) |
| 11 | 151,751 | 0.0012 (2.3e-5) | 0.0012 (2.2e-5) | 0.00084 (1.4e-5) | 0.0022 (6.2e-5) |
| 12 | 139,684 | 8.0e-4 (1.4e-5) | 0.00073 (1.2e-5) | 0.00049 (7.5e-6) | 0.0017 (4.8e-5) |
| 13 | 113,390 | 0.0010 (1.8e-5) | 0.00094 (1.6e-5) | 0.00061 (9.8e-6) | 0.0018 (4.9e-5) |
| 14 | 97,335 | 0.0011 (2.0e-5) | 0.0010 (1.8e-5) | 0.00065 (1.1e-5) | 0.0020 (5.6e-5) |
| 15 | 85,307 | 0.0010 (1.8e-5) | 0.00098 (1.7e-5) | 6.0e-4 (9.6e-6) | 0.0020 (5.8e-5) |
| 16 | 92,007 | 0.00088 (1.6e-5) | 0.00084 (1.5e-5) | 0.00054 (8.4e-6) | 0.0021 (6.2e-5) |
| 17 | 79,478 | 0.0012 (2.3e-5) | 0.0011 (2.0e-5) | 0.00069 (1.1e-5) | 0.0021 (6.0e-5) |
| 18 | 87,105 | 0.0010 (1.8e-5) | 0.00095 (1.7e-5) | 0.00058 (9.2e-6) | 0.0023 (6.8e-5) |
| 19 | 72,794 | 0.0012 (2.3e-05) | 0.0012 (2.1e-5) | 0.00082 (1.4e-5) | 0.0022 (6.2e-5) |
| 20 | 68,881 | 0.0014 (2.6e-5) | 0.0015 (2.7e-5) | 0.00078 (1.3e-5) | 0.0024 (7.0e-5) |
| 21 | 45,068 | 0.0018 (3.4e-5) | 0.0017 (3.2e-5) | 0.00098 (1.7e-5) | 0.0024 (7.1e-5) |
| 22 | 40,378 | 0.0016 (3.1e-5) | 0.0016 (2.9e-5) | 0.0010 (1.8e-5) | 0.0027 (8.1e-5) |

Each $\hat{\ell}_i$ and its standard error are in parentheses, as estimated in **Equation 7**.
SNP, single-nucleotide polymorphism.

when the missing rate is high, we suggest the genotype matrix should be imputed by other advanced imputation tools.

The most time-consuming part of X-LD was the construction of GRM $\mathbf{K} = \frac{1}{m}\widetilde{\mathbf{X}}\widetilde{\mathbf{X}}^T$, and the established computational time complex was $\mathcal{O}\left(n^2 m\right)$. However, if $\widetilde{\mathbf{X}}$ is decomposed into $\widetilde{\mathbf{X}} = \left[\widetilde{\mathbf{X}}_{[t_1,]} \vdots \widetilde{\mathbf{X}}_{[t_2,]} \vdots \cdots \vdots \widetilde{\mathbf{X}}_{[t_z,]}\right]$, in which $\widetilde{X}_{[t_i,]}$ has dimension of $n \times B$, using Mailman algorithm the computational time complex for building $\mathbf{K}$ can be reduced to $\mathcal{O}\left(\frac{n^2 m}{\log_3 m}\right)$ (**Liberty and Zucker, 2009**). This idea of embedding Mailman algorithm into certain high-throughput genomic studies has been successful, and our X-LD software is also leveraged by absorbing its recent practice in genetic application (**Wu and Sankararaman, 2018**).

## Results

### Statistical properties of the proposed method

*Table 1* introduces the symbols frequently cited in this study. As schematically illustrated in *Figure 1*, $\ell_g$ could be decomposed into $C$ $\ell_i$ and $\frac{C(C-1)}{2}$ unique $\ell_{i\cdot j}$ components. We compared the estimated

**Table 5.** LD decay regression analysis for 26 cohorts.

| Cohort (n) | LD-decay regression* | | | | Population parameters† | | | |
|---|---|---|---|---|---|---|---|---|
| | $\hat{b}_0$ | $\hat{b}_1$ | $R$ | LD decay score | $\bar{F}_{st}$ (%) | Ancestry | True LD ‡ |
| MSL (85) | 0.00041 | 29.97 | 0.84 | 0.0421 | 0.013 | AFR | 0.62727273 |
| GWD (113) | 0.00031 | 30.17 | 0.83 | 0.0439 | 0.009 | AFR | 0.65934066 |
| YRI (107) | 0.00030 | 30.64 | 0.85 | 0.0436 | 0.010 | AFR | 0.66292135 |
| ESN (99) | 0.00037 | 34.82 | 0.87 | 0.0436 | 0.011 | AFR | 0.65420561 |
| ACB (96) | 0.00053 | 39.62 | 0.88 | 0.0451 | 0.021 | AFR | 0.63194444 |
| LWK (99) | 0.00046 | 40.52 | 0.92 | 0.0447 | 0.014 | AFR | 0.64615385 |
| ASW (61) | 0.0015 | 46.88 | 0.83 | 0.0472 | 0.031 | AFR | 0.57142857 |
| CHS (105) | 0.00046 | 52.36 | 0.87 | 0.0555 | 0.010 | EA | 0.67375887 |
| CDX (93) | 0.00055 | 53.77 | 0.83 | 0.0557 | 0.012 | EA | 0.66666667 |
| KHV (99) | 0.00044 | 53.79 | 0.87 | 0.0560 | 0.011 | EA | 0.68345324 |
| CHB (103) | 0.00041 | 54.90 | 0.90 | 0.0558 | 0.010 | EA | 0.69402985 |
| JPT (104) | 0.00045 | 57.75 | 0.85 | 0.0568 | 0.010 | EA | 0.68965517 |
| BEB (86) | 0.00045 | 48.84 | 0.88 | 0.0556 | 0.012 | SA | 0.66911765 |
| ITU (102) | 0.00048 | 49.58 | 0.89 | 0.0546 | 0.016 | SA | 0.66433566 |
| STU (102) | 0.00055 | 52.84 | 0.89 | 0.0546 | 0.015 | SA | 0.64516129 |
| PJL (96) | 0.00054 | 54.00 | 0.90 | 0.0546 | 0.017 | SA | 0.67073171 |
| GIH (103) | 0.00057 | 55.81 | 0.91 | 0.0562 | 0.017 | SA | 0.65868263 |
| TSI (107) | 0.00041 | 53.17 | 0.91 | 0.0558 | 0.010 | EUR | 0.68939394 |
| IBS (107) | 0.00039 | 54.22 | 0.92 | 0.0555 | 0.010 | EUR | 0.7 |
| CEU (99) | 0.00045 | 54.23 | 0.89 | 0.0559 | 0.011 | EUR | 0.68085106 |
| GBR (91) | 0.00047 | 58.23 | 0.91 | 0.0555 | 0.012 | EUR | 0.68027211 |
| FIN (99) | 0.00054 | 59.24 | 0.86 | 0.0579 | 0.011 | EUR | 0.67073171 |
| MXL (64) | 0.0014 | 66.13 | 0.89 | 0.0558 | 0.036 | AMR | 0.6 |
| PUR (104) | 0.00059 | 67.20 | 0.89 | 0.0571 | 0.014 | AMR | 0.67039106 |
| CLM (94) | 0.00069 | 75.97 | 0.95 | 0.0572 | 0.017 | AMR | 0.66985646 |
| PEL (85) | 0.0012 | 78.15 | 0.85 | 0.0598 | 0.028 | AMR | 0.61290323 |
| 1KG (2503) | 0.0061 | 40.65 | 0.55 | | 0.066 | Mixed | 0.51587302 |

LD, linkage disequilibrium; SNP, single-nucleotide polymorphism.

*The regression intercept $b_0$ and the coefficients $b_1$ are as represented in **Equation 3**.

†The column for LD decay score was taken as the mean of the estimated $r^2 - \frac{1}{n}$ from PopLDdecay in a physical distance of 1500 kb, which was approximated to the area under the curve in **Figure 5A** for each cohort; $F_{st}$ was approximated by $\frac{\lambda_1}{n}$, in which $\lambda_1$ the largest eigenvalue for the cohort. $r^2$ was the estimated LD statistic from PLINK (--r2).

‡True LD is defined as $\frac{\hat{\ell}_{i\cdot j}}{\hat{\ell}_{i\cdot j} + \hat{b}_0}$.

$\ell_i$ and $\ell_{i\cdot j}$ in X-LD with those being estimated in PLINK (known as '--r2,' and the estimated squared Pearson's correlation LD is denoted as $r^2$). Considering the substantial computational cost of PLINK, only 100,000 randomly selected autosome SNPs were used for each 1KG cohort, and 22 $\hat{\ell}_i$ and 231 $\hat{\ell}_{i\cdot j}$ were estimated. After regressing 22 $\hat{\ell}_i$ against those of PLINK, we found that the regression slope was close to unity and bore an anticipated intercept a quantity of approximately $\frac{1}{n}$ (**Figure 2A and B**). In other words, PLINK gave $\frac{1}{n}$ even for SNPs of no LD. However, when regressing 231 $\hat{\ell}_{i\cdot j}$ estimates

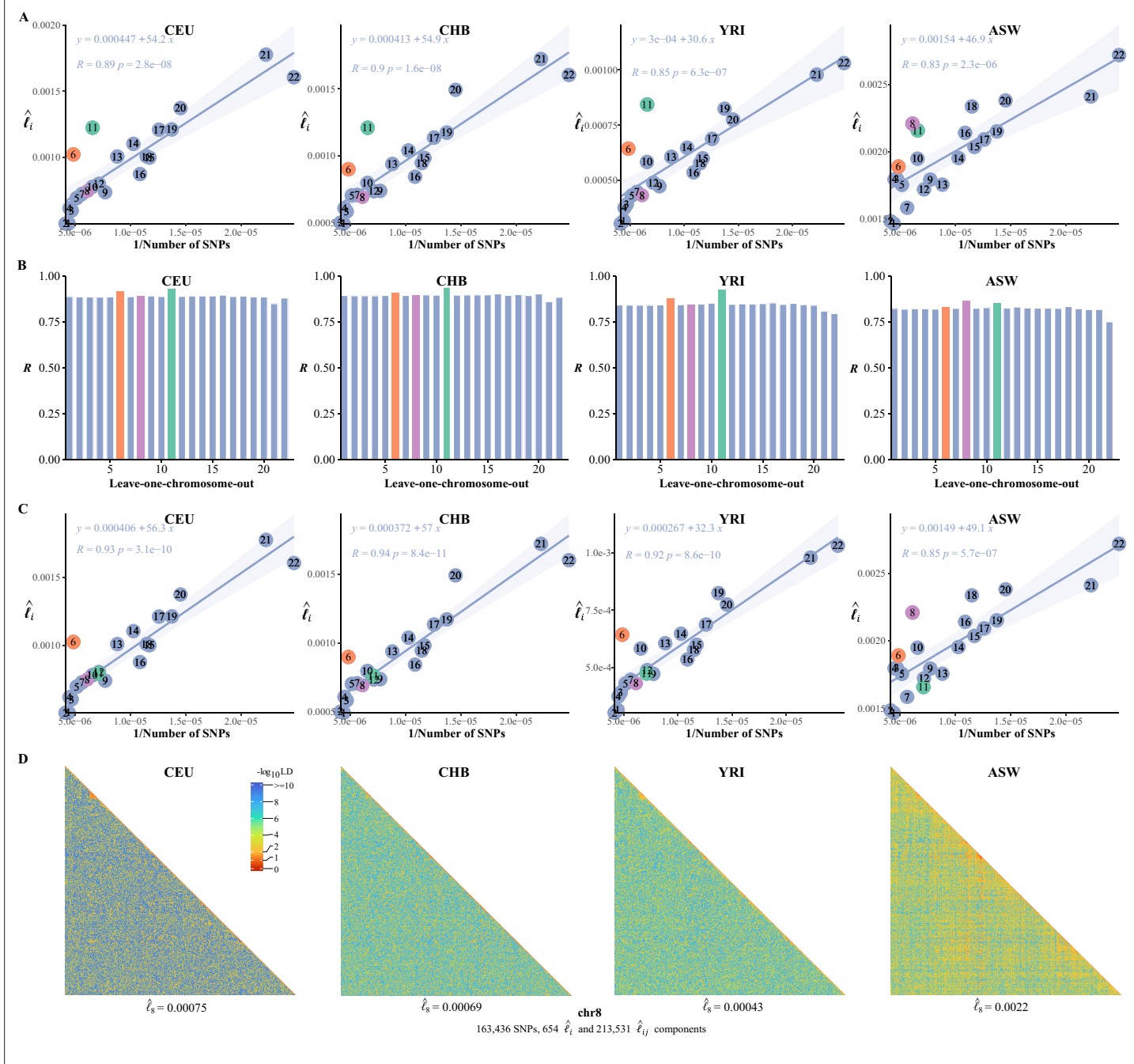

**Figure 6.** The correlation between the inversion of the single-nucleotide polymorphism (SNP) number and $\hat{\ell}_i$. (**A**) The correlation between the inversion of the SNP number and $\hat{\ell}_i$ in CEU, CHB, YRI, and ASW. (**B**) Leave-one-chromosome-out strategy is adopted to evaluate the contribution of a certain chromosome on the correlation between the inverse of the SNP number and $\hat{\ell}_i$. (**C**) The correlation between the inversion of the SNP number and chromosomal linkage disequilibrium (LD) in CEU, CHB, YRI, and ASW after removing the centromere region of chromosome 11. (**D**) High-resolution illustration for LD grids for chromosome 8 in CEU, CHB, YRI, and ASW. For each cohort, we partition chromosome 8 into consecutive LD grids (each LD grid contains 250 ×250 SNP pairs). For visualization purposes, LD is transformed to a -log10-scale, with smaller values (red hues) representing larger LD, and a value of 0 representing that all SNPs are in LD.

The online version of this article includes the following figure supplement(s) for figure 6:

**Figure supplement 1.** Influence of expanding of single-nucleotide polymorphism (SNP) numbers on the correlation between the inverse of the SNP number and chromosomal linkage disequilibrium (LD) in ASW.

against those of PLINK, it was found that largely because of the tiny quantity of $\hat{\ell}_{i \cdot j}$ it was slightly smaller than 1 but statistically insignificant from 1 in these 26 1KG cohorts (mean of 0.86 and SD of 0.10, and its 95% CI was (0.664, 1.056)); when the entire 1KG samples were used, its much larger LD due to subgroups, nearly no estimation bias was found (*Figure 2A and B*). In contrast, because of their much larger values, $\hat{\ell}_i$ components were always consistent with their corresponding estimates from PLINK (mean of 1.03 and SD of 0.012, 95% CI was (1.006, 1.053), bearing an ignorable bias). Furthermore, we also combined the African cohorts together (MSL, GWD, YRI, ESN, LWK, totaling 599 individuals), the East Asian cohorts together (CHS, CDX, KHV, CHB, and JPT, totaling 504 individuals), and the European cohorts together (TSI, IBS, CEU, GBR, and FIN, totaling 503 individuals), and the resemblance pattern between X-LD and PLINK was similar as observed in each cohort alone (*Figure 2—figure supplement 1*). The empirical data in 1KG verified that the proposed method was sufficiently accurate.

To fairly evaluate the computational efficiency of the proposed method, the benchmark comparison was conducted on the first chromosome of the entire 1KG dataset ($n = 2,503$ and $m = 225,967$), and 10 CPUs were used for multi-thread computing. Compared with PLINK, the calculation efficiency of X-LD was nearly 30–40 times faster for the tested chromosome, and its computational time of X-LD was proportional to $\mathcal{O}\left(\frac{n^2 m}{\log_3 m}\right)$ (*Figure 2—figure supplement 2*). So, X-LD provided a feasible and reliable estimation of large-scale complex LD patterns. More detailed computational time of the tested tasks is reported in their corresponding sections below; since each 1KG cohort had a sample size of around 100, otherwise specified the computational time was reported for CHB ($n = 103$) as a reference (*Table 2*). In order to test the capability of the software, the largest dataset tested was CONVERGE ($n = 10,640$, and $m = 5,215,820$), and it took 77,508.00 s, about 22 hr, to estimate 22 autosomal $\hat{\ell}_i$ and 231 $\hat{\ell}_{i \cdot j}$ (*Figure 1A*); when zooming into chromosome 2 of CONVERGE, on which 420,949 SNPs had been evenly split into 1000 blocks and yielded 1000 $\hat{\ell}_u$ grids, and 499,500 $\hat{\ell}_{uv}$ LD grids, it took 45,125.00 s, about 12.6 hr, to finish the task (*Figure 1B*).

## Ubiquitously extended LD and population structure/admixture

We partitioned the 2,997,635 SNPs into 22 autosomes (*Figure 3A*, *Figure 3—figure supplement 1*), and the general LD patterns were as illustrated for CEU, CHB, YRI, ASW, and 1KG. As expected, $\hat{\ell}_{i \cdot j} < \bar{\hat{\ell}}_g < \hat{\ell}_i$ for each cohort (*Figure 3B*). As observed in these 1KG cohorts, all three LD measures were associated with population structure, which was surrogated by $\bar{F}_{st} \approx \frac{\lambda_1}{n}$, and their squared correlation $R^2$ was greater than 0.8. ACB, ASW, PEL, and MXL, which all showed certain admixture, tended to have much greater $\hat{\ell}_g$, $\hat{\ell}_i$, and $\hat{\ell}_{i \cdot j}$ (*Table 3* and *Figure 3B*). In contrast, East Asian (EA) and European (EUR)-orientated cohorts, which showed little within-cohort genetic differentiation—as their largest eigenvalues were slightly greater than 1—had their aggregated LD relatively low and resembled each other (*Table 3*). Furthermore, for several European (TSI, IBS, and FIN) and East Asian (JPT) cohorts, the ratio between $\hat{\ell}_{i \cdot j}$ and $\hat{\ell}_i$ components could be smaller than 0.1, and the smallest ratio was found to be about 0.091 in FIN. The largest ratio was found in 1KG that $\hat{\ell}_{i \cdot j} = 5.7e - 3$ and $\hat{\ell}_i = 6.5e - 3$, and the ratio was 0.877 because of the inflated LD due to population structure. A more concise statistic to describe the ratio between $\ell_{i \cdot j}$ and $\ell_i$ was $\tilde{\ell}_{i \cdot j}$ (*Equation 8*), and the corresponding value for 231 scaled $\tilde{\ell}_{i \cdot j}$ for FIN was $\bar{\tilde{\ell}}_{i \cdot j} = 0.10$ (SD of 0.027) and for 1KG was $\bar{\tilde{\ell}}_{i \cdot j} = 0.88$ (SD of 0.028).

In terms of computational time, for 103 CHB samples, it took about 101.34 s to estimate 22 autosomal $\hat{\ell}_i$ and 231 $\hat{\ell}_{i \cdot j}$; for all 1KG 2503 samples, X-LD took about 3008.29 s (*Table 1*). Conventional methods took too long to complete the analyses in this section, so no comparable computational time was provided. For detailed 22 $\hat{\ell}_i$ and 231 $\hat{\ell}_{i \cdot j}$ estimates for each 1KG cohort, please refer to *Supplementary file 1* (Excel sheet 1–27).

## Detecting exceedingly high LD grids shaped by variable recombination rates

We further explored each autosome with high-resolution grid LD visualization. We set $\mathfrak{m} = 250$, so each grid had the $\ell_{uv}$ for $250 \times 250$ SNP pairs. The computational time complex was $\mathcal{O}\left(n^2\left(m_i + \frac{\beta_i^2}{4}\right)\right)$, in which $\beta_i = \frac{m_i}{250}$, and with our proposed method in CHB it cost 66.86 s for chromosome 2, which

had 241,241 SNPs and totaled 466,095 unique grids, and 3.22 s only for chromosome 22, which had 40,378 SNPs and totaled 13,203 unique grids (*Table 1*). In contrast, under conventional methods those LD grids were not very likely to be exhaustively surveyed because its computational cost was $\mathcal{O}\left(nm_i^2\right)$: for CHB chromosome 2, it would have taken about 40 hr as estimated. As the result was very similar for $\mathfrak{m} = 500$ (*Figure 4—figure supplement 1*), we only report the results under $\mathfrak{m} = 250$ below.

As expected, chromosome 6 (206,165 SNPs, totaling 340,725 unique grids) had its HLA cluster showing much higher LD than the rest of chromosome 6. In addition, we found a very dramatic variation of the HLA cluster LD $\hat{\ell}_{HLA}$ (28,477,797–33,448,354 bp, totaling 3160 unique grids) across ethnicities. For CEU, CHB, YRI, and ASW, their $\hat{\ell}_6 = 0.0010$, 0.00090, 0.00064, and 0.0019, respectively, but their corresponding HLA cluster grids had $\hat{\ell}_{HLA} = 0.042$, 0.029, 0.025, and 0.022, respectively (*Figure 4*). Consequently, the largest ratio for $\frac{\hat{\ell}_{HLA}}{\hat{\ell}_6}$ was 42.00 in CEU, 39.06 in YRI, and 32.22 in CHB, but was reduced to 11.58 in ASW. Before the release of CHM13 (*Hoyt et al., 2022*), chromosome 11 had the most completely sequenced centromere region, which had much rarer recombination events, and all four cohorts showed a strong LD $\hat{\ell}_{11.c}$ around the centromere (46,061,947–59,413,484 bp, totaling 1035 unique grids) regardless of their ethnicities (*Figure 4*). $\hat{\ell}_{11} = 0.0012$, 0.0012, 0.00084, and 0.0022, respectively, and $\hat{\ell}_{11.c} = 0.098$, 0.10, 0.079, and 0.094, respectively; the ratio for $\frac{\hat{\ell}_{11.c}}{\hat{\ell}_{11}} = 81.67$, 83,33, and 94,05, for CEU, CHB, and YRI, respectively; the lowest ratio was found in ASW of 42.73. In addition, removing the HLA region of chromosome 6 or the centromere region of chromosome 11 would significantly reduce $\hat{\ell}_6$ or $\hat{\ell}_{11}$ in comparison with the random removal of other regions (*Figure 4—figure supplement 2*).

## Model-based LD decay regression revealed LD composition

The real LD block size was not exact of $\mathfrak{m} = 250$ or $\mathfrak{m} = 500$, but an unknown parameter that should be inferred in computational intensive 'LD decay' analysis (*Zhang et al., 2019*; *Chang et al., 2015*). We conducted the conventional LD decay for the 26 1KG cohorts (*Figure 5A*), and the time cost was 1491.94 s for CHB. For each cohort, we took the area under the LD decay curve in the LD decay plot, and it quantified approximately the LD decay score for each cohort. The smallest score was 0.0421 for MSL, and the largest was 0.0598 for PEL (Table 5). However, this estimation did not take into account the real extent of LD, so it was not precise enough to reflect the LD decay score. For example, for admixture population, such as the American cohorts, the extent of LD would be longer.

In contrast, we proposed a model-based method, as given in *Equation 3*, which could estimate LD decay score (regression coefficient $b_1$) and long-distance LD score (intercept $b_0$) jointly. Given the estimated 22 $\hat{\ell}_i$ (*Supplementary file 1*; see *Table 4* for four representative cohorts and Supplementary R code), we regressed each autosomal $\hat{\ell}_i$ against its corresponding inversion of SNP number, and all yielded positive slopes (Pearson's correlation $\mathcal{R} > 0.80$, *Table 5* and *Figure 5B*), an observation that was consistent with genome-wide spread of recombination hotspots. This linear relationship could consequently be considered the norm for a relatively homogeneous population as observed in most 1KG cohorts (*Figure 5—figure supplement 1*), while for all the 2503 1KG samples $\mathfrak{R} = 0.55$ only (*Table 5*), indicating that the population structure and possible differentiated recombination hotspots across ethnicities disturbed the assumption underlying *Equation 3* and smeared the linearity. We extracted $b_0$ and $b_1$ for the 26 1KG cohorts for further analysis. The rates of LD decay score, as indicated by $b_1$, within the African cohorts (AFR) were significantly faster than the other continents, consistent with previous observation that the African population had relatively shorter LD *Gabriel et al., 2002*; while subgroups within the American continent (AMR) tended to have extended LD range due to their admixed genetic composition (*Table 4* and *Figure 5*). Notably, the correlation between $b_1$ and the approximated LD decay score was $\mathfrak{R} = 0.88$. The estimated $\bar{F}_{st}$ was highly correlated with $b_0$ ($\mathfrak{R} = 0.94$).

A common feature was universally relative high LD of chromosome 6 and 11 in the 26 1KG cohorts (*Figure 5—figure supplement 1*). We quantified the impact of chromosome 6 and 11 by leave-one-chromosome-out test in CEU, CHB, YRI, and ASW for details (*Figure 6A and B*) and found that dropping chromosome 6 off could lift $\mathfrak{R}$ on average by 0.017 and chromosome 11 by 0.046. One possible explanation was that the centromere regions of chromosomes 6 and 11 have been assembled more completely than other chromosomes before the completion of CHM13 (*Hoyt et al., 2022*), whereas meiotic recombination tended to be reduced around the centromeres (*Hinch et al., 2019*). We estimated $\ell_i$ after having knocked out the centromere region (46,061,947–59,413,484 bp, chr 11) in CEU,

CHB, YRI, and ASW, and chromosome 11 then did not deviate much from their respective fitted lines (**Figure 6C**). A notable exceptional pattern was found in ASW, chromosome 8 of which had even more deviation than chromosome 11 ($\mathfrak{R}$ was 0.83 and 0.87 with and without chromosome 8 in leave-one-chromosome out test) (**Figure 6B**). The deviation of chromosome 8 of ASW was consistent even more SNPs were added (**Figure 6—figure supplement 1**). We also provided high-resolution LD grids illustration for chromosome 8 (163,436 SNPs, totaling 214,185 grids) of the four representative cohorts for more detailed virtualization (**Figure 6D**). ASW had $\hat{\ell}_8 = 0.0022$, but 0.00075, 0.00069, and 0.00043 for CEU, CHB, and YRI, respectively.

## Discussion

In this study, we present a computationally efficient method to estimate the mean LD of genomic grids of many SNP pairs. Our LD analysis framework is based on GRM, which has been embedded in variance component analysis for complex traits and genomic selection (**Goddard, 2009**; **Visscher et al., 2014**; **Chen, 2014**). The key connection from GRM to LD is bridged via the transformation between $n \times n$ matrix and $m \times m$ matrix, in particular here via Isserlis's theorem under the fourth-order moment (**Isserlis, 1918**). With this connection, the computational cost for estimating the mean LD of $m \times m$ SNP pairs is reduced from $\mathcal{O}(nm^2)$ to $\mathcal{O}(n^2m)$, and the statistical properties of the proposed method are derived in theory and validated in 1KG datasets. In addition, as the genotype matrix $\mathbf{X}$ is of limited entries {0, 1, 2}, assuming missing genotypes are imputed first, using Mailman algorithm the computational cost of GRM can be further reduced to $\mathcal{O}\left(\frac{n^2m}{\log_3 m}\right)$ (**Liberty and Zucker, 2009**). The largest data tested so far for the proposed method has a sample size of 10,640 and more than 5 million SNPs, so it can complete genomic LD analysis in 77,508.00 s (**Table 1**). The weakness of the proposed method is obvious that the algorithm remains slow when the sample size is large or the grid resolution is increased. With the availability of such a UK Biobank data (**Bycroft et al., 2018**), the proposed method may not be adequate, and much advanced methods, such as randomized implementation for the proposed methods, are needed.

We also applied the proposed method into 1KG and revealed certain characteristics of the human genomes. Firstly, we found the ubiquitous existence of extended LD, which likely emerged because of population structure, even very slightly, and admixture history. We quantified the $\hat{\ell}_i$ and $\hat{\ell}_{i\cdot j}$ in 1KG, and as indicated by $\tilde{\ell}_{i\cdot j}$ we found that the inter-chromosomal LD was nearly an order lower than intra-chromosomal LD; for admixed cohorts, the ratio was much higher, even very close to each other such as in all 1KG samples. Secondly, variable recombination rates shaped peak of local LD. For example, the HLA region showed high LD in the European and East Asian cohorts, but relatively low LD in such as YRI, consistent with their much longer population history. Thirdly, there existed a general linear correlation between $\ell_i$ and the inversion of the SNP number, a long-anticipated result that is as predicted with genome-wide spread of recombination hotspots (**Hinch et al., 2019**). One outlier of this linear norm was chromosome 11, which had so far the most completely genotyped centromere and consequently had more elevated LD compared with other autosomes. We anticipate that with the release of CHM13 the linear correlation should be much closer to unity (**Hoyt et al., 2022**). Of note, under the variance component analysis for complex traits, it is often a positive correlation between the length of a chromosome (as surrogated by the number of SNPs) and the proportion of heritability explained (**Chen et al., 2014**).

In contrast, throughout the study recurrent outstanding observations were found in ASW. For example, in ASW the ratio of $\hat{\ell}_{HLA}/\hat{\ell}_6$ substantially dropped compared with that of CEU, CHB, or YRI as illustrated in **Figure 4**. Furthermore, chromosome 8 in ASW fluctuated upward most from the linear correlation (**Figure 6**) even after various analyses, such as expanding SNP numbers. One possible explanation may lie under the complex demographic history of ASW, which can be investigated and tested in additional African American samples or possible existence for epistatic fitness (**Ni et al., 2020**).

## Acknowledgements

GBC conceived and initiated the study. XH, TNZ, and YL conducted simulation and analyzed data. GBC, XH, TNZ, GAQ, and JZ developed the software. GBC wrote the first draft of the article. All authors contributed to the writing, discussion of the paper, and validation of the results. This work was

supported by the National Natural Science Foundation of China (31771392), 110202101032 (JY-09), and GZY-ZJ-KJ-23001. The funders played no role in the design, preparation, and submission of the article.

## Additional information

### Funding

| Funder | Grant reference number | Author |
|---|---|---|
| National Natural Science Foundation of China | 31771392 | Guo-Bo Chen |
| China National Tobacco Corporation | 110202101032(JY-09) | Guo-Bo Chen |

The funders had no role in study design, data collection and interpretation, or the decision to submit the work for publication.

### Author contributions

Xin Huang, Data curation, Software, Formal analysis, Validation, Visualization, Methodology, Writing – original draft; Tian-Neng Zhu, Data curation, Software, Formal analysis, Validation, Visualization, Methodology, Writing – review and editing; Ying-Chao Liu, Data curation, Formal analysis, Validation, Visualization; Guo-An Qi, Formal analysis, Validation; Jian-Nan Zhang, Software; Guo-Bo Chen, Conceptualization, Resources, Software, Formal analysis, Supervision, Validation, Methodology, Writing – original draft, Project administration, Writing – review and editing

### Author ORCIDs

Tian-Neng Zhu (ID) http://orcid.org/0009-0007-7507-4521
Guo-An Qi (ID) http://orcid.org/0000-0002-2412-3932
Guo-Bo Chen (ID) http://orcid.org/0000-0001-5475-8237

### Ethics

Human subjects: This study investigated variation in previously published anonymized genome data from the 1000 Genomes Project.

Joint Public Review: https://doi.org/10.7554/eLife.90636.3.sa1
Author Response https://doi.org/10.7554/eLife.90636.3.sa2

## Additional files

### Supplementary files
• MDAR checklist
• Supplementary file 1. Extended data for 1KG LD estimation.

### Data availability

Public genetic datasets used in this study can be freely downloaded from the following URLs.1000 Genomes Project: https://www.ebi.ac.uk/eva/?eva-study=PRJEB30460. CONVERGE: https://www.ebi.ac.uk/eva/?eva-study=PRJNA289433. All data generated or analysed during this study are included in the manuscript and supporting file. The CONVERGE dataset was used to generate Figure 1 and the other figures were generated from 1000 Genomes Project data.

The following previously published datasets were used:

| Author(s) | Year | Dataset title | Dataset URL | Database and Identifier |
|-----------|------|---------------|-------------|-------------------------|
| Lowy-Gallego E, Fairley S, Zheng-Bradley X, Ruffier M, Clarke L, Flicek P | 2019 | Variant calling on GRCh38 with the 1000 genomes samples | https://www.ebi.ac.uk/eva/?eva-study=PRJEB30460 | European Nucleotide Archive, PRJEB30460 |
| Na Cai, Bigdeli TB, Kretzschmar WW, Li Y, Liang J, Hu J, Peterson RE, Bacanu S, Webb BT, Riley B, Li Q, Marchini J, Mott R, Kendler KS, Flint J | 2017 | Study of Major Depression in Chinese women | https://www.ebi.ac.uk/eva/?eva-study=PRJNA289433 | European Nucleotide Archive, PRJNA289433 |

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
