## [Editor Report · eLife assessment]

This study presents a **useful** new approach for efficient computation of statistics on correlations between genetic variants (linkage disequilibrium, or LD), which the authors apply to quantify the extent of LD across chromosomes. The method and its derivation are **solid**. The authors document that cross-chromosome LD can be substantial, which has implications for geneticists who are interested in population structure and its impact on genetic association studies.

---

## [Referee Report · Joint Public Review]

Summary:

In this paper, the authors point out that the standard approach of estimating LD is inefficient for datasets with large numbers of SNPs, with a computational cost of O(nm^2), where n is the number of individuals and m is the number of SNPs. Using the known relationship between the LD matrix and the genomic-relatedness matrix, they can calculate the mean level of LD within the genome or across genomic segments with a computational cost of O(n^2m). Since in most datasets, n<

Strengths:

Generally, for computational papers like this, the proof is in the pudding, and the authors have been successful at their aim of producing an efficient computational tool. The most compelling evidence of this in the paper are Figure 2 and Supplementary Figure S2. In Figure 2, they report how well their X-LD estimates of LD compare to estimates based on the standard approach using PLINK. They appear to have very good agreement. In Figure S2, they report the computational runtime of X-LD vs PLINK, and as expected X-LD is faster than PLINK as long as it is evaluating LD for more than 8000 SNPs.

Weakness:

This method seems to be limited to calculating average levels of LD in broad regions of the genome. While it would be possible to make the regions more fine-grained, doing so appears to make this approach much less efficient. As such, applications of this method may be limited to those proposed in the paper, for questions where average LD of large chromosomal segments is informative.

Impact:

This approach seems to produce real gains for settings where broad average levels of LD are useful to know, but it will likely have less of an impact in settings where fine-grained levels are LD are necessary (e.g., accounting for LD in GWAS summary statistics).

---

## [Author Response]

The following is the authors’ response to the original reviews.

**Reviewer #1 (Public Review)**
Summary:Huang and colleagues present a method for approximation of linkage disequilibrium (LD) matrices. The problem of computing LD matrices is the problem of computing a correlation matrix. In the cases considered by the authors, the number of rows (n), corresponding to individuals, is small compared to the number of columns (m), corresponding to the number of variants. Computing the correlation matrix has cubic time complexity [O(nm2)], which is prohibitive for large samples. The authors approach this using three main strategies:1. they compute a coarsened approximation of the LD matrix by dividing the genome into variant-wise blocks which statistics are effectively averaged over;2. they use a trick to get the coarsened LD matrix from a coarsened genomic relatedness matrix (GRM), which, with O(n2m) time complexity, is faster when n << m;3. they use the Mailman algorithm to improve the speed of basic linear algebra operations by a factor of log(max(m,n)). The authors apply this approach to several datasets.Strengths:The authors demonstrate that their proposed method performs in line with theoretical explanations.The coarsened LD matrix is useful for describing global patterns of LD, which do not necessarily require variant-level resolution.They provide an open-source implementation of their software.Weaknesses:The coarsened LD matrix is of limited utility outside of analyzing macroscale LD characteristics.The method still essentially has cubic complexity--albeit the factors are smaller and Mailman reduces this appreciably. It would be interesting if the authors were able to apply randomized or iterative approaches to achieve more fundamental gains. The algorithm remains slow when n is large and/or the grid resolution is increased.

Thanks for your positive and accurate evaluation! We acknowledge the weakness and include some sentences in Discussion.

“The weakness of the proposed method is obvious that the algorithm remains slow when the sample size is large or the grid resolution is increased. With the availability of such as UK Biobank data (Bycroft et al., 2018), the proposed method may not be adequate, and much advanced methods, such as randomized implementation for the proposed methods, are needed.”

**Reviewer #2 (Public Review)**
Summary:In this paper, the authors point out that the standard approach of estimating LD is inefficient for datasets with large numbers of SNPs, with a computational cost of [O(nm2)], where n is the number of individuals and m is the number of SNPs. Using the known relationship between the LD matrix and the genomic- relatedness matrix, they can calculate the mean level of LD within the genome or across genomic segments with a computational cost of O(n2m). Since in most datasets, n<<m, this can lead to major computational improvements. They have produced software written in C++ to implement this algorithm, which they call X-LD. Using the output of their method, they estimate the LD decay and the mean extended LD for various subpopulations from the 1000 Genomes Project data.Strengths:Generally, for computational papers like this, the proof is in the pudding, and the authors appear to have been successful at their aim of producing an efficient computational tool. The most compelling evidence of this in the paper is Figure 2 and Supplementary Figure S2. In Figure 2, they report how well their X- LD estimates of LD compare to estimates based on the standard approach using PLINK. They appear to have very good agreement. In Figure S2, they report the computational runtime of X-LD vs PLINK, and as expected X-LD is faster than PLINK as long as it is evaluating LD for more than 8000 SNPs.Weakness:While the X-LD software appears to work well, I had a hard time following the manuscript enough to make a very good assessment of the work. This is partly because many parameters used are not defined clearly or at all in some cases. My best effort to intuit what the parameters meant often led me to find what appeared to be errors in their derivation. As a result, I am left worrying if the performance of X-LD is due to errors cancelling out in the particular setting they consider, making it potentially prone to errors when taken to different contexts.

Thanks for you critical reading and evaluation. We do feel apologize for typos, which have been corrected and clearly defined now (see Eq 1 and Table 1). In addition, we include more detailed mathematical steps, which explain how LD decay regression is constructed and consequently finds its interpretation (see the detailed derivation steps between Eq 3 and Eq 4).

Impact:I feel like there is value in the work that has been done here if there were more clarity in the writing. Currently, LD calculations are a costly step in tools like LD score regression and Bayesian prediction algorithms, so a more efficient way to conduct these calculations would be useful broadly. However, given the difficulty I had following the manuscript, I was not able to assess when the authors’ approach would be appropriate for an extension such as that.

See our replies below in responding to your more detailed questions.

**Reviewer #1 (Recommendations For The Authors)**
There are numerous linguistic errors throughout, making it challenging to read.It is unclear how the intercepts were chosen in Figure S2. Since theory only gives you the slopes, it seems like it would make more sense to choose the intercept such that it aligns with the empirical results in some way.

Thanks for your critical evaluation. We do feel apologize some typos, and we have read it through and clarify the text as much as possible. In addition, we included Table 1, which introduces mathematical symbols of the paper.

In Figure S2, the two algorithms being compared have different software implementations, PLINK vs X-LD. Their real performance not only depended on the time complexity of the algorithms (right-side y-axis), but also how the software was coded. PLINK is known for its excellent programming. If we could have programmed as well as Chris Chang, the performance of X-LD should have been even better and approach the ratio m/n. However, even under less skilled programming, X-LD outperformed plink.

**Reviewer #2 (Recommendations For The Authors):**
Thank you for the chance to review your manuscript. It looks like compelling work that could be improved by greater detail. Providing the level of detail necessary may require creating a Supplementary Note that does a lot of hand-holding for readers like me who are mathematically literate but who don’t have the background that you do. Then you can refer readers to the Supplement if they can’t follow your work.

We fix the problems and style issues as possible as we can.

Regarding the weakness section in the public review, here are a few examples of where I got confused, though this list is not exhaustive.1. Consider Equation 1 (line 100), which I believe must be incorrect. Imagine that g consists of two SNPs on different chromosomes with correlation rho. Then ell_g (which is defined as the average squared elements of the correlation matrix) would be

ell_g = 1/4 (1 + 1 + rho^2 + rho^2) = (1+rho^2)/2.

But ell_1=1 and ell_2=1 and ell_12=rho^2 (The average squared elements of the chromosome-specific correlation matrices and the cross-chromosome correlation matrix, respectively). So

sum(ell_i)+sum(ell_ij) = 1 + 1 + rho^2 + rho^2 = (1+rho^2)*2.

I believe your formulas would hold if you defined your LD values as the sum of squared correlations instead of the mean, but then I don’t know if the math in the subsequent sections holds. I think this problem also holds for Eq 2 and therefore makes Eqs 3 and 4 difficult to interpret.

Thanks for your attentive review and invaluable suggestions. We acknowledge the typo in calculating the mean in Eq 1, resulting in difficulties in understanding the equations. We sincerely apologize for this oversight. To address this issue and ensure clarity in the interpretation of Eq 3 and Eq 4, we have provided more detailed explanations (see the derivation between Eq 3 and Eq 4).

2. I didn’t know what the parameters are in Equation 3. The vector ell needs to be defined. Is it the vector of ell_i for each chromosomal segment i? I’m also confused by the definition of m_i, which is defined on line 113 as the “SNP number of the i-th chromosome.” Do the authors mean the number of SNPs on the i-th chromosomal segment? If so, it wasn’t clear to me how Eq 2 and Eq 3 imply Eq 4. Further, it wasn’t clear to me why E(b1) quantifies the average LD decay of the genome. I’m used to seeing plots of average LD as a function of distance between SNPs to calculate this, though I’m admittedly not a population geneticist, so maybe this is standard. Standard or not, readers deserve to have their hands held a bit more through this either in the text or in a Supplementary Note.

Thanks for your insightful feedback. When we were writing this paper, our actually focus was Eq 3 and to establish the relationship between chromosomal LD and the reciprocal of the length of chromosome (Fig 6A) – which was surrogated by the number of SNPs, the correlation between ell_i and 1/m_i.

We asked around our friends who are population geneticists, who anticipated the correlation between chromosomal LD (ell) and 1/m. The rationale simple if one knows the very basis of population genetics. A long chromosome experiences more recombination, which weakens LD for a pair of loci. In particular, for a pair of loci D_t=D_0 (1-c)^t. D_t the LD at the t generation, D_0 at the 0 generation, and c the recombination fraction. As recombination hotspots are nearly even distributed along the genome, such as reported by Science 2019;363:eaau8861, the chromosome will be broken into the shape in Author response image 1 (Fig 1C, newly added). Along the diagonal you see tight LD block, which will be vanished in the further as predicted by D_t equation, and any loci far away from each other will not be in LD otherwise raised by such as population structure. Ideally, we assume the diagonal block of aveage size of m×m and average LD of a SNP with other SNPs inside the diagonal block (red) is l_u; and, in contrast, off-diagonal average LD (light red) to be l_uv. This logic is hidden but employed in such as ld score regression and prs refinement using LD structure.

**Author response image 1. sa2fig1:** 

But, how to estimate chromosomal LD (ell), which is overwhelming [O(nm2)] as our friends said!So, the Figure 6A is logically anticipated by a seasoned population geneticist, but has never been realized because of [O(nm2)] is nightmare. Often, those signature patterns should have been employed as showcases in releasing new reference data, such as HapMap. However, to our knowledge, this signature linear relationship has never been illustrated in those reference data.

If you further test a population geneticist, if any chromosome will deviate from this line (Fig 6A)? The answer most likely will be chromosome 6 because of the LD tight HLA region. However, it is chromosome 11 because of its most completed sequenced centromere. Chr 11 is a surprise! With T2T sequenced population, Chr 11 will not deviate much. We predict!

However, we suspect whether people appreciate this point, we shift our focus to efficient computation of LD—which is more likely understood. We acknowledge the lack of clarity in notation definitions and the absence of the derivation for the interpretation of b1 and b0 for LD decay regression. So, we have added a table to provide an explanation of the notation (see the Table 1) and provided additional derivations, which explained how LD decay regression was derived (see the derivation between Eq 3 and Eq 4). Figure 1C provides illustration for the underlying assumption under LD.

The technique to bridge Eq 2~3 to Eq 4 is called “building interpretation”. It once was one of the kernel tasks for population genetics or statistical genetics, and a classical example is Haseman-Elston regression (Behavior Genetics, 1972, 2:3-19). When it is moving towards a data-driven style, the culture becomes “shut up, calculate”. Finding interpretation for a regression is a vanishing craftmanship, and people often end up with unclear results!

3. In line 135, it’s not clear to me what is meant by Gio2. If it is GioGio, then wouldn’t the resulting matrix be a matrix of zeros since Gio is zero everywhere except the lower off-diagonal? So maybe it is GioT∗Gio? But then later in that line, you say that the square of this matrix is the sum of several terms of the form Gk1k2. Are these the scalar elements of the G matrix? But then the sum is a scalar, which can’t be true since E(Gio2) is a matrix.

Thanks for your attentive review. We indeed confused the definition of matrices and their elements, and Gio should refer to the stacked off-diagonal elements of matrix Gi. So, Gio is a vector for variable gij – the relationship between sample i and j. We assume the reviewer use R software, then E(Gio2) corresponds to mean (G[row⁡(G)<col⁡(G)]∧2).

See the text between Eq 5 and Eq 6.

“We extract two vectors kio, which stacks the off-diagonal elements of Ki, and kid, which takes the diagonal elements of Ki.”

In addition, E(Gdiag )×n+E(Goff-diag )n(n−1)=0, so the ground truth is that E(Gio)=−1n−1, but not zero.

To clarify these math symbols, we replace G with K, so as to be consistent with our other works (see Table 1).

To derive the means and the sampling variances for ℓi and ℓi⋅j, the Eq 7 can be established by some modifications on the Delta method as exampled in Appendix I of Lynch and Walsh’s book (Lynch and Walsh, 1998). We added this sentence near Eq 7 in the main text.